# ZGS-Based Event-Driven Algorithms for Bayesian Optimization in Fully Distributed Multi-Agent Systems

## Abstract

*Bayesian optimization* (BO) is a well-established framework for globally optimizing expensive-to-evaluate black-box functions with impressive efficiency. Although numerous BO algorithms have been developed for the centralized machine learning setting and some recent works have extended BO to the tree-structured federated learning, no previous studies have investigated BO within a fully distributed *multi-agent system* (MAS) in the field of *distributed learning* (DL). Addressing this gap, we introduce and investigate a novel paradigm, *Distributed Bayesian Optimization* (DBO), in which agents cooperatively optimize the same costly-to-evaluate black-box objectives. An innovative generalized algorithm, *Zero-Gradient-Sum-Based Event-Driven Distributed Lower Confidence Bound* (ZGS-ED-DLCB), is proposed to overcome the significant challenges of DBO and DL: We (a) adopt a surrogate model based on random Fourier features as an approximate alternative to a typical Gaussian process to enable the exchange of local knowledge between neighboring agents, and (b) employ the event-driven mechanism to enhance communication efficiency in MASs. Moreover, we propose a novel generalized fully distributed convergence theorem, which represents a substantial theoretical and practical breakthrough wrt the ZGS-based DL. The performance of our proposed algorithm has been rigorously evaluated through theoretical analysis and extensive experiments, demonstrating substantial advantages over the state-of-the-art baselines.

## 1 Introduction

*Bayesian Optimization* (BO) has become a prominent tool for tackling problems that involve optimizing expensive-to-evaluate black-box functions, especially when resources are limited. In the field of *distributed learning* (DL), agents (specifically, *Internet of Things* (IoT) edge devices) aim to solve identical learning tasks through cooperative training with their one-hop neighbors, while avoiding the exchange of local and private raw data. Cooperative training in DL typically focuses on utilization of large datasets with low uncertainty and has made significant progress, as evidenced by numerous publications Koloskova et al. (2019a;b); Chen & Ren (2016); Scardapane et al. (2015); Aysal et al. (2008); Xiao et al. (2007); Lopes & Sayed (2008; 2007); Ai et al. (2016); Ren et al. (2018; 2020); Zhao & Chen (2021); Liu & Xie (2021). Specifically, the works Ai et al. (2016); Ren et al. (2018; 2020) present a set of synchronous DL algorithms based on the *zero gradient sum* (ZGS) method (Section 3.1), and the subsequent works Zhao & Chen (2021); Liu & Xie (2021); Chen & Ren (2016) further develop ZGS-based *event-driven* DL algorithms. However, traditional DL does not consider budget constraints for agents, while many practical optimization tasks that necessitate costly evaluations of black-box functions are constrained by limited budgets. Moreover, the strengths of BO can drive notable progress in distributed non-convex optimization and high-uncertainty distributed convex optimization (see, e.g., a recent review Yang et al. (2019)). These situations naturally prompt the integration of BO into DL, a novel paradigm, which we term *Distributed Bayesian Optimization* (DBO) (Section 2.2).

DBO has promising applications for real-world multi-agent scenarios, including cooperative hyperparameter tuning of multi-agent learning models (e.g., SVM (Dai et al., 2021; 2020), decision tree-based models (Li et al., 2020b;a) and DNN (McMahan et al., 2017)), collaborative chemical/material design (Zhang et al., 2020; Griffiths & Hernández-Lobato, 2020) and joint patient selection for medical

tests across multiple hospitals (Yu et al., 2015), etc. In hyperparameter optimization, confidentiality extends to hyperparameters; revealing them could expose underlying optimization strategies or other private information, potentially offering a competitive advantage. Nonetheless, privacy-preserving cooperation often outperforms isolated optimization. Unfortunately, despite the potential benefits in these applications, several challenges arise in DBO or impact the DL setting in general.

One challenge plagues DBO but not DL, due to the strict requirement to prevent local raw data exchange between neighboring agents for privacy protection. While DL also requires rigorous protection of an agent's privacy, the knowledge exchanged between neighboring agents, such as output weight parameters of local learning models, avoids the privacy risk associated with raw data flows. In BO, *Gaussian process* (GP) is the most popular choice for surrogate modeling of an objective function. However, a typical GP is nonparametric, leaving no parameter that can represent local GP models and be transferred to neighboring agents without infringing on their privacy, apart from the private raw data regarding local BO. Fortunately, *random Fourier features* (RFF) (Section 2.1) can be adopted as an approximation approach for a GP by establishing a linear model. The weight parameters of the RFF-based surrogate model reflect information of the approximated GP, and can be naturally exchanged between neighboring agents without private raw data flows. Moreover, the RFF-based distributed *lower confidence bound* (DLCB) functions (Section 2.2) can be utilized to derive promising predictions by trading-off exploitation and exploration under high uncertainty.

DBO is also essential to address the challenge of communication inefficiency and burden in DL. Although only output weight parameters are communicated in DL, real-time synchronous communication between neighboring agents demands considerable communication bandwidth and processing efficiency. However, bandwidth constraints limit the maximum communication frequency for each agent within a certain time interval, with increased frequency causing proportional energy consumption. To alleviate this, we incorporate an asynchronous *event-driven mechanism* (Section 3.2) into the DBO paradigm. The event-driven mechanism effectively reduces excessive resource consumption during simultaneous communication, thus enhancing training efficiency and accelerating algorithm convergence rate in DBO/DL.

The recent works Sim et al. (2021); Dai et al. (2021; 2020) have extended BO to the tree-structured federated learning, pioneering multi-agent BO with privacy protection. Specifically, the works Dai et al. (2021; 2020) propose novel *federated BO* (FBO) algorithms and the work Sim et al. (2021) presents a *collaborative BO* (CBO) algorithm that assumes the presence of a centralized mediator. Despite practical implementations of FBO and CBO algorithms incorporating event-driven updates, these updates lack corresponding theoretical analysis. In addition, the aforementioned traditional DL works Koloskova et al. (2019a;b); Chen & Ren (2016); Scardapane et al. (2015); Aysal et al. (2008); Xiao et al. (2007); Lopes & Sayed (2008; 2007); Ai et al. (2016); Ren et al. (2018; 2020); Zhao & Chen (2021); Liu & Xie (2021) do not consider the expensive black-box optimization problems utilizing limited data. Addressing these significant gaps, we aim to design a DBO algorithm that solves costly-to-evaluate black-box optimization problems in a fully distributed *multi-agent system* (MAS) is nontrivial and promising, given the lack of previous work investigating BO in fully distributed MASs within the DL field.

Building upon the discussions above, we present a novel asynchronous DBO algorithm, *ZGS-based event-driven DLCB* (ZGS-ED-DLCB), aimed at addressing the expensive-to-evaluate black-box optimization problems in fully distributed MASs. We rigorously evaluate the performance of ZGS-ED-DLCB through both theoretical analysis and extensive experiments. To the best of our knowledge, the main contributions of our paper are summarized as follows:

- We propose a novel generalized *distributed learning* (DL) paradigm, termed *Distributed Bayesian Optimization* (DBO), which represents a natural and significant extension to the field of DL.

- Our work presents the first attempt to tackle two significant challenges in DL: 1) solving expensive-to-evaluate black-box optimization problems, and 2) tackling high-uncertainty DL problems, which arise when dealing with limited data.

- It is the first time to provide: 1) theoretical treatment for the event-driven mechanism in privacy-preserving multi-agent BO, and 2) the theoretical convergence analysis of multi-agent BO algorithms with the event-driven mechanism.

- We propose a novel generalized theorem based on the Laplacian spectral radius of a fully distributed MAS. This algorithm represents a substantial theoretical and practical breakthrough wrt the ZGS-

based DL Zhao & Chen (2021); Chen & Ren (2016); Ai et al. (2016); Ren et al. (2018; 2020) and guarantees the rigorous global consensus convergence of ZGS-ED-DLCB (Section 4.1).

## 2 PRELIMINARIES AND BACKGROUND

### 2.1 ALTERNATIVE TO A GAUSSIAN PORCESS (GP): RANDOM FOURIER FEATURES (RFF)-BASED SURROGATES

BO is a popular framework for optimizing expensive-to-evaluate black-box functions through limited observations. A Gaussian process (GP, see definition in App. B)has become a prominent surrogate model in BO during the last thirty years (Cressie, 1990; Jiang et al., 2020). However, a GP is a non-parametric model, resulting in no local information except private raw data can be exchanged between neighboring agents and hence violating agents' data privacy in DL. Thus, in this paper, according to Bochner's theorem (see App. O), RFFs are adopted as an approximate approach for a GP's kernel function by utilizing $M$-dimensional ($M \in \mathbb{Z}_+$) random features, i.e., $k(\mathbf{x}, \mathbf{x}') = \varphi(\mathbf{x})^\top \varphi(\mathbf{x}')$ (Dai et al., 2021; 2020; Rahimi et al., 2007). The RFF approximation method has theoretical guarantees on performance with high probability, i.e., $\sup_{\mathbf{x}, \mathbf{x}' \in \mathcal{D}} |k(\mathbf{x}, \mathbf{x}') - \varphi(\mathbf{x})^\top \varphi(\mathbf{x}')| \leqslant \varepsilon, \varepsilon \triangleq \mathcal{O}(M^{-1/2})$. An RFF approximation on a GP can be regarded as a linear surrogate model, i.e., $f(\mathbf{x}) \triangleq \varphi(\mathbf{x})^\top w$. The weight vector $w$ contains the information about the original GP surrogate and can also be exchanged between neighboring agents in the DL setting. RFFs have been employed in the FL-based BO frameworks (Dai et al., 2021; 2020). App. B illustrates the approximation ability of the RFF-based surrogate.

### 2.2 DISTRIBUTED BAYESIAN OPTIMIZATION (DBO)

**Distributed Lower Confidence Bound (DLCB).** Inspired by the *Upper Confidence Bound* (UCB) algorithm wrt the *multi-arm bandit* (MAB) problem, the GP-UCB algorithm (Bogunovic et al., 2016; Kandasamy et al., 2015; Srinivas et al., 2012; 2009) predicts the potential optimal locations when solving expensive maximization problems. GP-UCB is transformed into GP-LCB if the problem is minimizing a black-box objective function. As aforementioned discussions, *RFF-based distributed LCB* (DLCB) is employed as the local acquisition function to trade off between local exploitation (i.e., latest RFF-based belief from $m_{i,t-1}(\mathbf{x})$) and local exploration (i.e., uncertainty from $\sigma_{i,t-1}(\mathbf{x})$) at the $t$th iteration step and further bounding the corresponding *global cumulative regret* $R_T$ (defined in (2)) in our work. The DLCB acquisition function is defined as:

$$\mathbf{x}_{i,t} = \min_{\mathbf{x} \in \mathcal{D}} m_{i,t-1}(\mathbf{x}) - \sqrt{\vartheta_{i,t-1}} \sigma_{i,t-1}(\mathbf{x}), \tag{1}$$

where $\sqrt{\vartheta_{i,t}} = c_{i,1} \log(c_{i,2}t)$ is tunable, balancing exploration and exploitation at the iteration step $t$. The regret bound is $R_T = \tilde{\mathcal{O}}(\max\{\sqrt{T}, T\epsilon^{1/6}\})$ wrt the SE kernel (Bogunovic et al., 2016).

**Notions of Regret in DBO.** A common objective of BO algorithms is the minimization of *global cumulative regret*, which can be extended and defined in the DL setting as:

$$R_T \triangleq \sum_{i=1}^{N} \sum_{t=1}^{T} [f(x_t^i) - f(x^*)], \tag{2}$$

in which $f(x_t^i) - f(x^*)$ is the *local instantaneous regret* of the $t$th iteration step, $\sum_{t=1}^{T} [f(x_t^i) - f(x^*)]$ is the *local cumulative regret* over the first $T$ iteration steps and $\sum_{i=1}^{N} [f(x_t^i) - f(x^*)]$ is the *global instantaneous regret* of the $t$th iteration step. A DBO algorithm will achieve eventual convergence to a global optimum (e.g., minimum in this paper) when it achieves *no regret* asymptotically, i.e., $\lim_{T \to \infty} R_T/T = 0$. The *global simple regret* $S_T \triangleq \sum_{i=1}^{N} \min_{t \in [T]} [f(x_t^i) - f(x^*)] \leqslant R_T/T$ takes the *local simple regret* $S_T^i \triangleq \min_{t \in [T]} [f(x_t^i) - f(x^*)]$ as sub-items. Thus, *no regret* is asymptotically achieved if $R_T$ increases sub-linearly or $S_T$ gradually converges to 0 equivalently.

**Problem Formulation.** In this paper, we introduce a novel distributed expensive black-box optimization problem over an $N$-agent system, whose characteristics are illustrated in App. D and where insufficient private raw data samples are locally stored at each agent. The entire dataset in a fully distributed MAS of $N$ agents is denoted as $\mathcal{S} = \cup_{i=1}^{N} \mathcal{S}_i$ of size $M = \sum_{i=1}^{N} N_i$. $\mathcal{S}_i = \{(x_i^l, y_i^l)\}_{l=1}^{N_i}$ of size $N_i$ is a private dataset and $(x_i^l, y_i^l)$ is an input-output pair wrt agent $i$. $X_i = [x_i^1, x_i^2, ..., x_i^{N_i}]^\top \in \mathbb{R}^{N_i}$

and $Y_i = [y_i^1, y_i^2, ..., y_i^{N_i}]^\top \in \mathbb{R}^{N_i}$ are private input and output vectors, respectively. $y_i^l$ is an evaluated output of input $x_i^l$ from a common expensive black-box function $f$, which is given by:

$$y_i^l = f(x_i^l) + \varepsilon_i^l, \tag{3}$$

in which $\varepsilon_i^l \in \mathbb{R}$ is noise, $l \in [N_i]$. (For brevity of denotation, $[N]$ is used to denote $\{1, ..., N\}$, $N \in \mathbb{Z}$.) And $f(x_i^l)$ can be formed as a linear RFF combination:

$$f(x_i^l) = \sum_{j=1}^n w_{ij} s_j(x_i^l) = s(x_i^l)^\top W_i, \tag{4}$$

where $s(x_i^l) = [s_1(x_i^l), s_2(x_i^l), ..., s_n(x_i^l)]^\top \in \mathbb{R}^n$ is an $n$-dimensional RFF vector and $W_i = [w_{i1}, w_{i2}, ..., w_{in}]^\top \in \mathbb{R}^n$ is a local output weight vector. Further, the vector form of $f(x_i^l)$ is:

$$F_i = S_i W_i, \tag{5}$$

in which $S_i = [s(x_i^1), s(x_i^2), ..., s(x_i^{N_i})]^\top \in \mathbb{R}^{N_i \times n}$ and $F_i = [f(x_i^1), f(x_i^2), ..., f(x_i^{N_i})]^\top \in \mathbb{R}^{N_i}$. (5) is the transformation basis for (6) and (7) in **Problem Transformation**. In contrast to the original (5), the transformed (6) and (7) are the global objective functions in DL and DBO, respectively.

**Problem Transformation.** We aim to present a novel DBO algorithm to solve the distributed expensive black-box problem. In DL, the machine learning problem is typically formulated as:

$$\min_{\mathcal{W}} G(\mathcal{W}) = \sum_{i=1}^N g(W_i) = \frac{1}{2} \sum_{i=1}^N (\|Y_i - S_i W_i\|^2 + \sigma \|W_i\|^2), \tag{6}$$

where $\mathcal{W} = \{W_1, W_2, ..., W_N\}$ is the set of local output weight vectors $W_i$, $G(\mathcal{W})$ is the global objective function and $g(W_i) = \frac{1}{2}(\|Y_i - S_i W_i\|^2 + \sigma \|W_i\|^2)$ is the local objective wrt agent $i$. $\sigma > 0$ is tunable and treated as a tradeoff between the 2-norms of local learning error and $W_i$. Obviously, the minimization of (6) is a quadratic strongly convex optimization problem. According to Conclusion 1 (see App. K), there exists a unique $W^*$ in theory making $G(\mathcal{W})$ reach its global minimum.

In the setting of DBO, the problem (6) can be extended into the following time-varying form as:

$$\min_{\mathcal{W}_t} G(\mathcal{W}_t) = \sum_{i=1}^N g(W_t^i) = \frac{1}{2} \sum_{i=1}^N (\|Y_t^i - S_t^i W_t^i\|^2 + \sigma \|W_t^i\|^2). \tag{7}$$

The typical (6) can be regarded as a one-iteration-step particular case of the extended (7). Thus, a DBO problem can be regarded as an extended version of a DL problem. Similarly, there exists a unique $W_t^*$ in theory making $G(\mathcal{W}_t)$ reach its global minimum wrt the $t$th iteration step.

**Remark 1.** At the $t$th iteration step of a DBO algorithm, the local objective function $g(W_t^i)$ in (7) is strongly convex and twice continuously differentiable.

## 3 ZERO-GRADIENT-SUM-BASED EVENT-DRIVEN DISTRIBUTED LCB (ZGS-ED-DLCB)

### 3.1 ZERO GRADIENT SUM (ZGS)

**ZGS strategy.** ZGS Lu & Tang (2012) is a distributed optimization strategy forcing the gradient sum of local objectives to be constant zero vector during sub-iterations of the $t$th iteration step, i.e.:

$$\sum_{i=1}^N \nabla g(W_t^i(k)) = \mathbf{0}_n. \tag{8}$$

**ZGS-based initialization.** At the $t$th iteration step of a DBO algorithm, $W_t^i(0)$, the initialization of $W_t^i(k)$, is forced to satisfy the ZGS strategy, i.e.:

$$\sum_{i=1}^N \nabla g(W_t^i(0)) = \mathbf{0}_n. \tag{9}$$

**Zero-gradient-difference-sum (ZGDS) strategy.** We design ZGDS, a novel **discrete variant** of the ZGS strategy demonstrating the ZGS property maintenance of $g(W_t^i)$ in a DBO algorithm, as follows:

$$\sum_{i=1}^N \left[ \nabla g(W_t^i(k+1)) - \nabla g(W_t^i(k)) \right] = \mathbf{0}_n. \tag{10}$$

**Lemma 1.** In the problem (7), a DBO algorithm satisfies the ZGS strategy iff it satisfies both the ZGS-based initialization and the ZGDS strategy (See App. F for the proof).

## 3.2 EVENT-DRIVEN MECHANISM

In the setting of DBO, the core of the asynchronous event-driven mechanism is the decentralized trigger function, which is defined as:

$$H_t^i(k) = \|e_t^i(k)\|^2 - \alpha\beta_t^k, \ k \in \mathbb{Z}, \tag{11}$$

where $0 < \beta_t < 1$, $\alpha > 0$ and the error variables $e_t^i(k) = \hat{W}_t^i(k) - W_t^i(k)$. $\hat{W}_t^i(0) = W_t^i(0)$, and thus, $e_t^i(0) = \mathbf{0}_n$. The event-driven mechanism avoids the **Zeno** behaviour Yu & Chen (2020); Chen & Ren (2016); Lamperski & Ames (2012), i.e., occurring infinite executions in a finite time period, which is detailedly illustrated in App. E.

In the iteration step $t$, agent $i$ monitors its own learned knowledge $W_t^i(k)$ at the $k$th sub-iteration step. And the current knowledge $W_t^i(k)$ is transferred to its neighboring agents as soon as the trigger function $H_t^i(k) > 0$ is satisfied. The latest transferred knowledge of agent $i$ is denoted as $\hat{W}_t^i(k) = W_t^i(k_{t,i}^{m_i})$, $k \in [k_{t,i}^{m_i}, k_{t,i}^{m_i+1})$, and its latest received knowledge from neighbors is given as $\hat{W}_t^j(k) = W_t^j(k_{t,j}^{m_j})$, $j \in \mathcal{N}_i$, where $k_{t,i}^{m_i}, k_{t,j}^{m_j} \in [0, \infty)$ respectively represent the trigger-times for agents $i$ and $j$ with $m_i, m_j = 0, 1, 2, \ldots$. The sequence of trigger-times for agent $i$ is defined iteratively as $k_{t,i}^{m_i+1} = \inf\{k : k > k_{t,i}^{m_i}, H_t^i(k) > 0\}$ with $0 \leqslant k_{t,i}^0 \leqslant k_{t,i}^1 \leqslant k_{t,i}^2 \leqslant \ldots$, in which $k_{t,i}^0$ is the first trigger-time.

**Remark 2.** Different from Chen & Ren (2016), all the parameters are time-varying with the iteration steps and hence more flexible in the DBO/DL setting.

## 3.3 ZGS-ED-DLCB ALGORITHM DESCRIPTION

In this paper, a novel resource-saving asynchronous DBO algorithm, i.e., ZGS-ED-DLCB, is proposed for solving the expensive black-box problem by employing the event-driven mechanism in the DL setting. In terms of the $N$ cooperative agents at the $t$th iteration step, $t \in [T]$, the ZGS-ED-DLCB algorithm is designed in the following two stages:

**Stage 1: Achieve global consensus convergence**

$$\begin{cases} W_t^i(k+1) = W_t^i(k) + \gamma_t[S_t^{i\top}S_t^i + \sigma I_{nd}]^{-1}\Big[\sum_{j \in \mathcal{N}_i} a_{ij}\big(\hat{W}_t^j(k) - \hat{W}_t^i(k)\big)\Big], \\ \hat{W}_t^i(k) = \hat{W}_t^i(k_{t,i}^{m_i}), k_{t,i}^{m_i} = k, \text{ if } H_t^i(k) \geqslant 0, k \in [k_{t,i}^{m_i}, k_{t,i}^{m_i+1}) \subseteq [1, K], K \in \mathbb{N}, \\ W_t^{i*} = W_t^i(K+1), \\ W_t^i(0) = [S_t^{i\top}S_t^i + \sigma I_{nd}]^{-1}S_t^{i\top}Y_t^i, \\ \hat{W}_t^i(0) = W_t^i(0) \text{ with } k_{t,i}^0 = 0, \end{cases} \tag{12}$$

in which $\gamma_t > 0$ is a gain parameter and $W_t^i$, initialized as $W_t^i(0)$, is obtained from the $t$th iteration step of ZGS-ED-DLCB. Moreover, (12) can be rewritten as the following matrix form:

$$\begin{cases} W_t(k+1) = W_t(k) - \gamma_t[S_t^\top S_t + \sigma I_{Nnd}]^{-1}(\mathcal{L} \otimes I_n)\big(W_t(k) + e_t(k)\big), \\ W_t^{opt} = W_t(K+1), k \in [1, K], K \in \mathbb{N}, \\ W_t(0) = [S_t^\top S_t + \sigma I_{Nnd}]^{-1}S_t^\top Y_t, \end{cases} \tag{13}$$

where $S_t = \text{diag}\{S_t^1, S_t^2, ..., S_t^N\} \in \mathbb{R}^{N(N_i+t-1) \times Nnd}$, $W_t(k) = [W_t^1(k)^\top, W_t^2(k)^\top, ..., W_t^N(k)^\top]^\top$, $W_t^{opt} = [W_t^{1*\top}, W_t^{2*\top}, ..., W_t^{N*\top}]^\top$ and $e_t(k) = [e_t^1(k)^\top, e_t^2(k)^\top, ..., e_t^N(k)^\top]^\top \in \mathbb{R}^{Nnd \times 1}$.

**Stage 2: Find potential observation locations using RFF-based DLCB**

Based on the obtained $W_t^{i*}$, each agent calculates the aforementioned mean $m_t(\mathbf{x}^*)$ and prediction variance $\hat{\sigma}^2(\mathbf{x}^*)$ at the iteration step $t$. Using the calculated $m_t(\mathbf{x}^*)$ and $\hat{\sigma}^2(\mathbf{x}^*)$, the RFF-based DLCB acquisition function is employed to calculate the next local observing locations, i.e., $\mathbf{x}_t^i$, from the common local noisy expensive black-box functions. Then calculate the *average global cumulative regret* $R_t/t$ at the $t$th iteration step and proceed to the iteration step $t + 1$.

To illustrate our ZGS-ED-DLCB algorithm more detailedly, App. C shows the algorithm pseudocode.

# 4 THEORETICAL RESULTS

In the problem (7), the positive definite matrix $S_t^{i\top} S_t^i + \sigma I_{nd}$ is the Hessian matrix of the local objective function $g(\mathcal{W}_t^i)$ wrt $W_t^i$, whose inverse controls the relative change rate of distinct components in the total cooperation from neighboring agents, i.e., $\sum_{j\in\mathcal{N}_i} a_{ij}\big(\hat{W}_t^j(k) - \hat{W}_t^i(k)\big)$, and further influences the local weight vector difference, i.e., $W_t^i(k+1) - W_t^i(k)$. The largest and smallest eigenvalues of the Hessian matrix are $\Xi_t^i = \lambda_{\max}(S_t^{i\top} S_t^i + \sigma I_{nd})$ and $\xi_t^i = \lambda_{\min}(S_t^{i\top} S_t^i + \sigma I_{nd})$, respectively. In the global perspective, $\Xi_t = \max_{i\in\mathcal{V}} \Xi_t^i$ and $\xi_t = \min_{i\in\mathcal{V}} \xi_t^i$.

The aforementioned eigenvalues of the Hessian matrix and the Laplacian matrix determine the convergence condition of ZGS-ED-DLCB jointly. Two properties on the convergence of ZGS-ED-DLCB are given as follows.

## 4.1 CONVERGENCE ANALYSIS OF ZGS-ED-DLCB

Main results concerning the rigorous convergence of our ZGS-ED-DLCB are given as follows.

**Connectivity Status Assumptions.** MAS network topologies in this work are assumed to be fixed, undirected and connected. Each agent is assumed to derive the Laplacian matrix $\mathcal{L}$ (or adjacency matrix $\mathcal{A}$, see App. D) of the *multi-agent system* (MAS) before the iterations of ZGS-ED-DLCB.

**Lemma 2.** Consider the Stage 1 (12) with the event-driven mechanism (11) under the undirected connected MAS network topology, if $\frac{\xi_t}{2\lambda_{\max}(\mathcal{L})} < \gamma_t < \min\big\{\frac{\Xi_t}{2\lambda_2(\mathcal{L})}, \frac{\xi_t(2\epsilon+\lambda_{\max}(\mathcal{L}))}{2\lambda_{\max}(\mathcal{L})(\epsilon+\lambda_{\max}(\mathcal{L}))}\big\}$ and $\epsilon > 0$, the convergence of agents in each iteration step can be illustrated by the following inequality:

$$\sum_{i=1}^{N} \|W_t^* - W_t^i(k)\|^2 \leqslant \frac{2}{\xi_t}\big[(V_t(0) - \rho_t)\theta_t^k + \rho_t\beta_t^k\big], \tag{14}$$

where $\rho_t = \frac{N\alpha\eta_2}{\beta_t - \theta_t}$, $\theta_t = 1 - \frac{2\lambda_2(\mathcal{L})\gamma_t\eta_1}{\Xi_t}$, $\xi_t = \min_{i\in[N]} \xi_t^i = \lambda_{\min}(S_t^\top S_t + \sigma I_{Nnd})$ with $\xi_t^i = \lambda_{\min}(S_t^{i\top} S_t^i + \sigma I_{nd})$, $\Xi_t = \max_{i\in[N]} \Xi_t^i$ with $\Xi_t^i = \lambda_{\max}(S_t^{i\top} S_t^i + \sigma I_{nd})$, $\eta_1 = 1 - \frac{\gamma_t\lambda_{\max}(\mathcal{L})^3\epsilon}{\xi_t^2} - \frac{\gamma_t\lambda_{\max}(\mathcal{L})}{\xi_t} - \frac{\lambda_{\max}(\mathcal{L})}{2\epsilon}$, $\eta_2 = \frac{\gamma_t^2\lambda_{\max}(\mathcal{L})^2}{\xi_t} + \frac{\gamma_t^2}{\epsilon} + \frac{\gamma_t\epsilon}{2}$ and $V_t(0) = \frac{1}{2}\sum_{i=1}^{N}\big(W_t^{i*} - W_t^i(0)\big)^\top(S_t^{i\top} S_t^i + \sigma I_{nd})\big(W_t^{i*} - W_t^i(0)\big)$. $\square$

App. G illustrates the proof of Lemma 2. From Lemma 2, we can derive that the ZGS-ED-DLCB algorithm is established an asymptotic convergence to the *global consensus*, i.e., $\lim_{k\to\infty} W_t^i(k) = W_t^{i*} = W_t^*$, after $t$ iteration steps. Moreover, there are $K$ sub-iteration steps in every iteration step. Empirically, when $K$ is no less than 500, $W_t^i$ of the ZGS-ED-DLCB and DCL algorithms converge to their corresponding *global consensus* at the $t$th iteration step, respectively. In the $t$th iteration step, $V_t(0)$ is the initial value of the designed global Lyapunov function $V_t(k)$, which is also defined in App. G. We can derive that a necessary condition for convergence is $0 < \gamma_t < \frac{\Xi_t}{2\lambda_2(\mathcal{L})}$, which indicates that $\gamma_t$ have to be restricted smaller than the ratio of $\Xi_t$ and $2\lambda_2(\mathcal{L})$. Furthermore, higher connectivity of an MAS results in larger $\lambda_2(\mathcal{L})$s and smaller ranges of $\gamma_t$.

**Lemma 3.** In theory, $\sigma$ is the infimum (or the greatest lower bound) for $\xi_t$ and $\xi_t^i$ in Lemma 2.

App. I illustrates the proof of Lemma 3. Empirically, $\xi_t$ usually equals to $\sigma$ constantly (e.g., when $M = 200$), $\lambda_{\max}(\mathcal{L})$ is a constant number and $\frac{\Xi_t}{2\lambda_2(\mathcal{L})}$ is far larger than $\frac{\xi_t(2\epsilon+\lambda_{\max}(\mathcal{L}))}{2\lambda_{\max}(\mathcal{L})(\epsilon+\lambda_{\max}(\mathcal{L}))}$ if $\epsilon$ is set appropriately. Thus, based on the practical experience, we can derive the following Theorem 1 wrt the convergence of ZGS-ED-DLCB. And Theorem 1 is the main theoretical result in our work.

**Theorem 1.** Consider the Stage 1 (12) with the event-driven mechanism (11) under the undirected connected MAS network topology, if $\frac{1}{2\lambda_{\max}(\mathcal{L})} < \gamma_t < \frac{3}{4\lambda_{\max}(\mathcal{L})}$ ($\xi_t = \sigma = 1, \epsilon = \lambda_{\max}(\mathcal{L})$), then the convergence of agents in each iteration step can be illustrated according to (15) as:

$$\sum_{i=1}^{N} \|W_t^* - W_t^i(k)\|^2 \leqslant 2\big[(V_t(0) - \rho_t)\theta_t^k + \rho_t\beta_t^k\big]. \tag{15}$$

The proof of Theorem 1 refers to that of Lemma 2 in App. G. In this paper, we set $\gamma_t = \frac{5}{8\lambda_{\max}(\mathcal{L})}$. From the perspective of theory and practical meaningfulness, the convergence of ZGS-ED-DLCB at each iteration step only relies on $\lambda_{\max}(\mathcal{L})$, i.e., the Laplacian spectral radius of the MAS. Moreover,

the setting of $\gamma_t$ is efficient to configure in agents and overcomes the drawback exist in the recent ZGS-based DL algorithms Chen & Ren (2016); Ren et al. (2018; 2020) theoretically and practically that all agents require the $S_t^i$ from the other agents to work. Such drawback means that the previous ZGS-based DL algorithms are not classical fully distributed algorithms and limit the practical feasibility and privacy protection of distinct agents.

**Remark 3: ZGS maintenance. (a) ZGS-based initialization.** At the iteration step $t$, $W_t^i(0)$ in the ZGS-ED-DLCB algorithm is designed as $[S_t^{i\top}S_t^i + \sigma I_{nd}]^{-1}S_t^{i\top}Y_t^i$, which is the initialization of $W_t^i(k)$, satisfies the ZGS-based initialization scheme, i.e., $\sum_{i=1}^N \nabla g\big(W_t^i(0)\big) = \mathbf{0}_n$. Moreover, $[S_t^{i\top}S_t^i + \sigma I_{nd}]^{-1}S_t^{i\top}Y_t^i$ is the minimizer of the local objective function $g(W_t^i)$ (Yang et al., 2019). **(b) ZGDS maintenance**. It is formulated as a recurrence relation derived from the Stage 1 that $\sum_{i=1}^N \big[\nabla g\big(W_t^i(k+1)\big) - \nabla g\big(W_t^i(k)\big)\big] = \sum_{i=1}^N \nabla g\big(W_t^i(k+1)\big) - \sum_{i=1}^N \nabla g\big(W_t^i(k)\big) = \sum_{i=1}^N (S_t^{i\top}S_t^i + \sigma I_{nd})\big(W_t^i(k+1) - W_t^i(k)\big) = \gamma_t \sum_{i=1}^N \sum_{j\in\mathcal{N}_i} a_{ij}\big(\hat{W}_t^j(k) - \hat{W}_t^i(k)\big) = \mathbf{0}_n$. Due to (a), the ZGS strategy can be constantly achieved for all $k$ in the $t$th iteration step, i.e., $\sum_{i=1}^N \nabla g\big(W_t^i(k)\big) = \mathbf{0}_n$. Then according to Lemma 2, the ZGS strategy can be constantly achieved for all $k$ in the $t$th iteration step, i.e., $\sum_{i=1}^N \nabla g\big(W_t^i(k)\big) = \mathbf{0}_n$, on the basis of (a) and (b).

**Remark 4: Fair Regret Based on Event-Driven Global Consensus.** ZGS-ED-DLCB ensures absolute fairness at each iteration by promoting *global consensus*. This is achieved as all agents indirectly process the entire dataset via local $W_t^i(k)$ exchanges, yielding identical learning results, i.e., $W_t^*$, in each iteration step. Each data point is equally significant in predicting potential $\mathbf{x}_t^i$ using sequential $W_t^*$ iterations, ensuring equal rewards for all agents. Additionally, the event-driven mechanism asynchronously fosters $W_t^i(k)$ updates while preserving the original $W_t^*$.

# 5 EXPERIMENTS AND DISCUSSIONS

In this section, we experimentally verify the performance and robustness of ZGS-ED-DLCB and comparing ZGS-ED-DLCB to the relevant baselines. In the comparisons, costly training of each algorithm is implemented conducted within the same limited iteration count.

## 5.1 EXPERIMENTAL SETTINGS

**Use Cases.** We select use cases covering synthetic (Section 5.2) and real-world (Section 5.3) experiments. Three 1D functions with many local minima in the synthetic experiments and two popular datasets from the UCI machine learning repository Dua & Graff (2017); Sigillito et al. (1989) in the real-world experiments are selected to investigate the resource-saving capability and performance improvement of ZGS-ED-DLCB over the baselines.

**Baselines.** Several existing DL baselines such as DCL Ai et al. (2016); Ren et al. (2018; 2020), ADMM-based Scardapane et al. (2015), ATC LMS and CTA LMS Cattivelli & Sayed (2009; 2008); Lopes & Sayed (2007; 2008) are considered.

**Performance Metrics.** The *global cumulative regret* $R_T$ (defined in (2)) is employed as the performance metric for comparisons wrt synthetic experiments in Sec. 5.2. We also use the performance of *misclassification rate* (MCR) wrt real-world experiments in Sec. 5.3. To clearly illustrate the resource-saving property of the event-driven mechanism in experiments of the both types, the metrics of triggering frequency and triggering instants are adopted.

## 5.2 SYNTHETIC EXPERIMENTS OF OPTIMIZING BENCHMARK FUNCTIONS WITH MANY LOCAL MINIMA

In the synthetic experiments, we consider 5 groups and each group owns 1 private agent. In the input region [-10, 10], each agent is randomly assigned with 10 noise-free data pairs as initialization. The computationally expensive cooperative training is limited in 5 iterations as budgets wrt the synthetic experiments. Each experiment is repeated 5 independent trials and the corresponding results are averaged over these trials.

**Metrics on Convergence wrt the MAS Topology.** The experiments are used to verify the performance of the ZGS-ED-DLCB algorithm over 5 agents on the 4 different MAS topologies, i.e., *path*, *ring*, *random* and *complete* (or *fully-connected*). Tab. 1 lists the determinant of convergence $\lambda_{\max}(\mathcal{L})$ and the metrics influencing the convergence rate, i.e., $\bar{d}$ and $\lambda_2(\mathcal{L})$.

**Remark 5.** In each iteration step of all the DL algorithms, $\lambda_2(\mathcal{L})$ and $\bar{d}$ influence the convergence rate jointly. $\lambda_2(\mathcal{L})$ represents the **algebraic connectivity** of a distributed MAS network topology, and $\bar{d}$ reflects the **geometrical connectivity**. For two different topologies, if the value of one metric is equal, the higher value of the other metric leads to faster global consensus convergence at each iteration step.

Table 1: Topology information of MASs with $N = 5$.

| Topology | $\lambda_2(\mathcal{L})$ | $\lambda_{\max}(\mathcal{L})$ | Average Degree $\bar{d}$ |
|---|---|---|---|
| *path* | 0.3820 | 3.6180 | 1.6 |
| *ring* | 1.3820 | 3.6180 | 2 (constant) |
| *random* | 1.3820 | 4.6180 | 2.4 |
| *complete* | 5 (or $N$) | 5 (or $N$) | 4 (or $N-1$) |

## 5.3 REAL-WORLD EXPERIMENTS

In real-world experiments, we study the performance advantages of ZGS-ED-DLCB over the baselines only in fully-connected MAS topologies because that the ADMM-based algorithm can only be implemented in complete topologies and the corresponding results are sufficiently representative due to the synthetic experimental results (see Tab. 2). Consider the data volume separated at agents and practical tight budgets, the cooperative training is limited in only a budget of 3 iteration steps. All the algorithms in each experiment are repeated 5 independent trials and the corresponding results are the averages of these trials.

**Classification of Iris Plants.** The dataset consisting of data samples from three classes of iris plants with four input attributes are separated at 10 agents. Each agent aims to cooperatively tune the hyperparameter C (ranging in [0.01, 1] and controlling the regularization strength) of a *logistic regression* (LR) for globally predicting the types of iris plants accurately by the 3 sequential observations.

**Classification of Radar Returns from the Ionosphere.** The dataset includes radar data collected in Goose Bay. 5 agents are involved and each agent are randolearningy assigned 70/71 radar data samples containing 34-dimensional

Table 2: $R_T/T$ per trial for the benchmark problems with many local minima wrt MAS topologies: (a) *Path*, (b) *Ring*, (c) *Random*, (d) *Complete*.

| Algorithm | $R_T/T$ per trial | | | |
|---|---|---|---|---|
| | Levy | Ackley | Griewank | |
| ZGS-ED-DLCB | **0.111** | **2.474** | **0.168** | |
| DCL | 0.233 | 4.319 | 0.674 | *path* |
| ATC-LMS | 23.021 | 39.779 | 0.862 | |
| CTA-LMS | 15.215 | 42.185 | 1.480 | |
| ZGS-ED-DLCB | **0.498** | **4.497** | **0.173** | |
| DCL | 0.605 | 6.800 | 0.759 | *ring* |
| ATC-LMS | 21.263 | 38.986 | 0.839 | |
| CTA-LMS | 10.404 | 41.433 | 1.035 | |
| ZGS-ED-DLCB | **0.182** | **4.780** | **0.357** | |
| DCL | 0.253 | 7.188 | 0.739 | *random* |
| ATC-LMS | 23.020 | 38.903 | 1.781 | |
| CTA-LMS | 13.840 | 42.803 | 1.530 | |
| ZGS-ED-DLCB | **0.101** | **4.114** | **0.626** | |
| DCL | 0.197 | 8.066 | 0.881 | |
| ATC-LMS | 20.567 | 38.700 | 1.870 | *complete* |
| CTA-LMS | 18.027 | 39.684 | 0.975 | |
| ADMM-based | 36.902 | 79.261 | 6.780 | |

continuous features and a binary label representing whether the radar returns are "good" or "bad". "Good" returns show evidence of some structure type in the ionosphere, while "Bad" ones do not. Each agent attempts to tune 2 hyperparameters (i.e., regularization parameter in [0.01, 2] and RBF kernel parameter in [0.01, 5]) of a classical SVM classifier cooperatively to train a globally precise radar return classifier through the 3 successive evaluations.

## 5.4 EXPERIMENTAL RESULTS

The synthetic comparison results are summarized in Tab. 2–3. Tab. 3 illustrates the comparison results of the average global cumulative regret $R_T/T$ and average triggering frequency per trial between ZGS-ED-DLCB and the baselines. Our ZGS-ED-DLCB totally outperforms the baselines in all MAS topologies of the experiments significantly wrt the metric $R_T/T$. Tab. 3 shows that ZGS-ED-DLCB benefits from the employed event-drvien mechanism and consumes substantially

less triggering frequency (approximately 200 vs 1000 for *path* and *random*, approximate 320 vs 1000 for *ring* and approximate 350 vs for *complete*) to achieve the global consensus of searching locations for the costly-to-evaluate black-box objective functions wrt the average triggering frequency metric.

Tab. 4–5 summarize the real-world comparison results. Tab. 4 shows that ZGS-ED-DLCB markedly reduces the average MCRs per trial in contrast to the tested baselines. It means that more appropriate hyperparameters of local SVM classifiers have been derived cooperatively to guarantee considerably fewer MCRs. Similar to the synthetic experiments, Tab. 5 also illustrates that ZGS-ED-DLCB achieves drastically lower consumption of triggering frequency during the iterations in contrast to the baselines wrt the *complete* MAS topology.

It can be found that ZGS-ED-DLCB saves more than 60% of the communication resource and the corresponding computation resource for each agent and meanwhile guarantees efficient global consensus convergence and better learning performance in contrast to the test baselines wrt the synthetic and real-world experiments.

Table 3: Average triggering frequency comparisons between ZGS-ED-DLCB and the baselines per trial wrt the benchmark problems.

| Topology | Algorithm | Avg triggering frequency per trial | | |
|---|---|---|---|---|
| | | Levy | Ackley | Griewank |
| *path* | ZGS-ED-DLCB | **206.28** | **200.44** | **204.4** |
| | Baselines | 1000 | 1000 | 1000 |
| *ring* | ZGS-ED-DLCB | **322.56** | **325.64** | **322.64** |
| | Baselines | 1000 | 1000 | 1000 |
| *random* | ZGS-ED-DLCB | **218.24** | **220.44** | **228.48** |
| | Baselines | 1000 | 1000 | 1000 |
| *complete* | ZGS-ED-DLCB | **355.20** | **356.04** | **354.84** |
| | Baselines | 1000 | 1000 | 1000 |

Due to space constraints, some experimental details and supplementary experiments are elaborated in App. P. Main supplementary experiments are briefly summarized as follows. In order to offer a more comprehensive elucidation of the asynchrony, sparsity, and efficiency in inter-agent communication under the event-driven mechanism of ZGS-ED-DLCB wrt the experiments, line charts illustrating the triggering frequency throughout iterations for all trials are presented. To further show the details, triggering instants at a randomly selected iteration step per trial are depicted based on the line charts.

Table 4: Average MCRs per trial for the real-world problems with wrt the *complete* topology of an MAS.

| Algorithm | Average MCRs per trial | |
|---|---|---|
| | Ionosphere | Iris |
| ZGS-ED-DLCB | **2.11%** | **8.80%** |
| DCL | 11.45% | 34.67% |
| ATC-LMS | 11.45% | 28.80% |
| CTA-LMS | 11.45% | 16% |
| ADMM-based | 35.91% | 28.40% |

*complete*

Table 5: Average triggering frequency comparisons between ZGS-ED-DLCB and the baselines per trial wrt the real-world problems.

| Topology | Algorithm | Avg triggering frequency per trial | |
|---|---|---|---|
| | | Ionosphere | Iris |
| **complete** | **ZGS-ED-DLCB** | **355.20** | **142.76** |
| | **Baselines** | 1000 | 1000 |

## 6 CONCLUSION

In this paper, we introduce the first DBO algorithm called ZGS-ED-DLCB to tackle the challenges of optimizing expensive black-box functions in fully distributed MASs. Our proposed algorithm leverages a surrogate model based on RFF, which serves as an approximate alternative to the traditional GP, enabling efficient exchange of local knowledge between neighboring agents. Moreover, we introduce the event-driven mechanism to enhance communication efficiency in MASs and establish a refined fully distributed convergence theorem based on the Laplacian spectral radius of an MAS, guaranteeing the rigorous global consensus convergence of ZGS-ED-DLCB. Extensive theoretical analysis and thorough experimental evaluations demonstrate the superior performance and significant advantages of our proposed algorithm compared to state-of-the-art algorithms. While our algorithm achieves promising results on the unconstrained optimization problems, further exploration is required for constrained distributed expensive black-box optimization problems.

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

## A    NOTATIONS

Throughout this paper, $\mathbb{R}$, $\mathbb{Z}_+$ and $\mathbb{N}$, respectively, represent the real number set, the positive integer set and the natural number set; $\mathbb{R}^n$ represents the set of $n$-dimensional column vectors with real elements; $\mathbb{R}^{n \times n}$ represents the set of $n \times n$ matrices with all real elements; $I_n$ represents the $n \times n$ identity matrix; $\mathbf{0}_n$ is an $n$-dimensional vector with all zeros; $\mathbf{1}_n$ is an $n$-dimensional vector with all ones; $A^\top$ is the transpose of matrix $A$; $\otimes$ represents the Kronecker product operation; $\| \cdot \|$ represents the Euclidean norm of a vector/matrix; $\rho(\cdot)$, $\lambda_{\min}(\cdot)$ and $\lambda_{\max}(\cdot)$ respectively represent the spectral radius, the minimum eigenvalue and the maximum eigenvalue of a matrix; $E(f)$ represents the expected value of a function $f$; $\nabla f$ and $\nabla^2 f$ represent the gradient and the Hessian matrix of $f$, respectively.

## B    GAUSSIAN PROCESS (GP) AND APPROXIMATION ABILITY OF RFF-BASED SURROGATE MODEL (DAI ET AL., 2020)

### B.1    GAUSSIAN PROCESS (GP)

A GP is denoted as $\mathcal{GP}(m(\mathbf{x}), k(\mathbf{x}, \mathbf{x}'))$ and specified by its mean function $m(\mathbf{x}) = E[f(\mathbf{x})]$ and covariance/kernel function $k(\mathbf{x}, \mathbf{x}') = cov[f(\mathbf{x}), f(\mathbf{x}')]$, for all $\mathbf{x}, \mathbf{x}' \in \mathcal{D} \subset \mathcal{R}^d$. Assumptions are wlog made that $m_t(\mathbf{x}) = 0$ and $k(\mathbf{x}, \mathbf{x}') \leqslant 1$, respectively. We mainly considers the popular squared exponential (SE) covariance function. In terms of a set of $s_t$ historical observations $D_{s_t} = \{(\mathbf{x}_1, f(\mathbf{x}_1)), ..., (\mathbf{x}_{s_t}, f(\mathbf{x}_{s_t}))\}$, the prediction mean $m_{s_t}(\mathbf{x})$ and prediction variance $\sigma_{s_t}^2(\mathbf{x})$ are respectively given as:

$$m_{s_t}(\mathbf{x}) \triangleq k_{s_t}(\mathbf{x})^\top [K_{s_t} + \Gamma^2 I]^{-1} y_{s_t}, \tag{16}$$

$$\sigma_{s_t}^2(\mathbf{x}, \mathbf{x}') \triangleq k(\mathbf{x}, \mathbf{x}') - k_{s_t}(\mathbf{x})^\top [K_{s_t} + \Gamma^2 I]^{-1} k_{s_t}(\mathbf{x}'), \tag{17}$$

where $\Gamma > 0$ is a regularization factor, $k_{s_t}(\mathbf{x}) = [k(\mathbf{x}, \mathbf{x}_{t'})]_{t' \in [s_t]}^\top$, $K_{s_t} = [k(\mathbf{x}_{t'}, \mathbf{x}_{t''})]_{t', t'' \in [s_t]}$ and $y_{s_t} = [y(\mathbf{x}_1), \dots, y(\mathbf{x}_{s_t})]^\top$, where $s_t$ is the available training dataset size at the iteration step $t$.

### B.2    APPROXIMATION ABILITY OF RFF-BASED SURROGATE MODEL

RFF-based surrogate model can be treated as a Bayesian linear model, whose approximation ability is detailedly illustrated in Appendices A and B of the previous work Dai et al. (2020). Refer to Dai et al. (2020) for more details.

## C    PSEUDOCODE OF ZGS-ED-DLCB

## D    ALGEBRAIC GRAPH THEORY

In our paper, a fully distributed multi-agent system (MAS) is modeled by an undirected connected graph $\mathcal{G} \triangleq \{\mathcal{V}, \mathcal{E}, \mathcal{A}\}$ composed of an $N$-agent set $\mathcal{V} = \{1, 2, \cdots, N\}$; an edge set $\mathcal{E} \subseteq \mathcal{V} \times \mathcal{V}$ denoting the communication routing; and a weighted adjacency matrix corresponding to $\mathcal{E}$, i.e., $\mathcal{A} = [a_{ij}] \in \mathbb{R}^{N \times N}$ with $a_{ij} \geqslant 0$ and $a_{ij} = a_{ji}$. For simplicity, we use $i$ as shorthand for agent $i$. An edge $e_{ij} = (i, j)$ in $\mathcal{G}$ is denoted as an unordered agent pair; $e_{ij} \in \mathcal{E}$ iff local knowledge exchanges exist between the agent neighbors $i$ and $j$, and $a_{ij} > 0$, $e_{ij} \in \mathcal{E} \Leftrightarrow e_{ji} \in \mathcal{E}$. In this paper, a neighbor means a one-hop neighbor. $a_{ii} = 0$ means there is no self loop of each agent since it is not self-communicated. In terms of the graph $\mathcal{G}$ and its complete graph $\bar{\mathcal{G}}$, the Laplacian matrix is defined as $\mathcal{L} = [l_{ij}] \in \mathbb{R}^{N \times N}$ and $\bar{\mathcal{L}}$, where $l_{ii} = \sum_{j=1}^{N} a_{ij}$ and $l_{ij} = -a_{ij}$ if $i \neq j$. The neighbor set of $i$ is defined as $\mathcal{N}_i = \{j \in \mathcal{V} \mid e_{ij} \in \mathcal{E}\}$. Moreover, $\mathcal{L}$ is positive semidefinite since $\mathcal{L} = \mathcal{K} \times \mathcal{K}^\top$, where $\mathcal{K}$ is the incidence matrix of an arbitrary orientation wrt $\mathcal{L}$ (Godsil & Royle, 2001). Hence, $\lambda_1(\mathcal{L}), \lambda_2(\mathcal{L}), \cdots, \lambda_N(\mathcal{L})$, the $N$ eigenvalues of $\mathcal{L}$, are totally nonnegative. Further assume that $\lambda_{\min}(\mathcal{L}) = \lambda_1(\mathcal{L}) \leqslant \lambda_2(\mathcal{L}) \leqslant \cdots \leqslant \lambda_N(\mathcal{L}) = \lambda_{\max}(\mathcal{L})$. Meanwhile, $\mathcal{L} \times \mathbf{1}_N = \mathbf{0}_n \Rightarrow \lambda_1(\mathcal{L}) = 0$. The number of zero eigenvalues wrt $\mathcal{L}$ equals the number of connected components wrt the graph $\mathcal{G}$. As $\mathcal{G}$ is connected, $\lambda_2(\mathcal{L})$ is the smallest of all nonzero eigenvalues,

---

**Algorithm 1** ZGS-ED-DLCB

---

**Input:** $N$ agents with their insufficient local initial training set $\mathcal{D}_1^i = \{(x_i^l, y_i^l)\}_{l=1}^{N_i}$, $i \in [N]$.
$X_1^i = [x_i^1, x_i^2, ..., x_i^{N_i}]^\top$, $Y_1^i = [y_i^1, y_i^2, ..., y_i^{N_i}]^\top$ and iteration number $T \in \mathbb{Z}_+$.
**for** $t = 1$ to $T$ **do**
 **Stage 1** ——————————————————————————————
 **for** each agent $i$ **in synchronous parallel do**
  $S_t^i \longleftarrow [\sqrt{2/M} \cos(s^\top X_t^i + b)]$
  $W_t^i(0) \longleftarrow [S_t^{i\top} S_t^i + \sigma I_{nd}]^{-1} S_t^{i\top} Y_t^i$
  $\hat{W}_t^i(0) \longleftarrow W_t^i(0)$
  $\hat{e}_t^i(0) \longleftarrow 0$
  $k_{t,i}^0 \longleftarrow 0$
 **end for**
 **for** $k = 0$ **to** $K$ **do**
  **for** each agent $i$ **in synchronous parallel do**
   $\hat{W}_t^i(k) \longleftarrow \hat{W}_t^i(k_{t,i}^{m_i})$
   $e_t^i(k) \longleftarrow \hat{W}_t^i(k) - W_t^i(k)$
   **if** $H_t^i(k) \geqslant 0$ do **then**
    $k_{t,i}^{m_i} \longleftarrow k$ **(asynchronous parallel)**
   **end if**
   $W_t^i(k+1) \longleftarrow W_t^i(k) + \gamma_t [S_t^{i\top} S_t^i + \sigma I_{nd}]^{-1} \Big[ \sum_{j \in \mathcal{N}_i} a_{ij} \big( \hat{W}_t^j(k) - \hat{W}_t^i(k) \big) \Big]$
  **end for**
 **end for**
 **Stage 2** ——————————————————————————————
 **for** each agent $i$ **in synchronous parallel do**
  Calculate $m_t(X_i)$ and $\hat{\sigma}^2(X_i)$
  Calculate local RFF-based LCB functions to derive the next local evaluation points $\mathbf{x}_t^i$ from
  local RFF-based surrogate models
  Query $\mathbf{x}_t^i$ to derive $\mathbf{y}_t^i$ from identical local black-box objective function $f$ and then update
  $D_{t+1}^i$ with $(X_{t+1}^i, Y_{t+1}^i)$ for the iteration step $t+1$
 **end for**
 Calculate $R_t/t$ **(synthetic experiment verification)**
**end for**
**Output:** $W_{\text{ZGS-ED-DLCB}}^* = W_T^{i*}$, $W_t^{i*}$ and $R_t/t$, $t \in [T]$.

---

i.e., $0 = \lambda_1(\mathcal{L}) < \lambda_2(\mathcal{L}) \leqslant \cdots \leqslant \lambda_N(\mathcal{L})$. Further the multiplicity of eigenvalue $N$ of $\bar{\mathcal{L}}$ is $N-1$, i.e., $\lambda_2(\bar{\mathcal{L}}) = \lambda_3(\bar{\mathcal{L}}) = \cdots = \lambda_N(\bar{\mathcal{L}}) = N$ Fiedler (1973).

# E   Zeno Behaviour in the DL setting

**Definition:** In the DL setting of our work, Zeno behavior occurs if triggering instants of agent $i$, i.e., $\{k_{t,i}^{m_i}\}_{m_i=0}^\infty$, satisfy that

$$\lim_{m_i \to \infty} k_{t,i}^{m_i} = \sum_{m_i=0}^\infty (k_{t,i}^{m_i} - k_{t,i}^{m_i-1}) \leqslant \tau_{t,i} < \infty$$

with a finite Zeno instant $\tau_{t,i} \leqslant 0$ and $k_{t,i}^0 = 0$.

This definition reveals the aforementioned phenomenon of Zeno behaviour that infinite number of executions occur in a finite time period. Our ZGS-ED-DLCB algorithm naturally avoids the Zeno behaviour of agents due to the resource-saving sampling scheme based on the designed local triggering functions $H_t^i(k)$ (11).

## F    PROOF OF LEMMA 1

**a. Necessity.** When $k = 0$, the ZGS strategy (8) turns into the ZGS-based initialization (9). Due to the ZGS strategy (8), we can derive that:

$$\sum_{i=1}^{N} \nabla g\big(W_t^i(1)\big) - \sum_{i=1}^{N} \nabla g\big(W_t^i(0)\big) = \sum_{i=1}^{N} [\nabla g\big(W_t^i(1)\big) - \nabla g\big(W_t^i(0)\big)] = \mathbf{0}_n,$$

$$\sum_{i=1}^{N} \nabla g\big(W_t^i(2)\big) - \sum_{i=1}^{N} \nabla g\big(W_t^i(1)\big) = \sum_{i=1}^{N} [\nabla g\big(W_t^i(2)\big) - \nabla g\big(W_t^i(1)\big)] = \mathbf{0}_n,$$

$$\vdots$$

$$\sum_{i=1}^{N} \nabla g\big(W_t^i(k)\big) - \sum_{i=1}^{N} \nabla g\big(W_t^i(k-1)\big) = \sum_{i=1}^{N} [\nabla g\big(W_t^i(k)\big) - \nabla g\big(W_t^i(k-1)\big)] = \mathbf{0}_n, \quad (18)$$

Thus, the ZGDS strategy (10) is constantly guaranteed over the sub-iterations in an iteration step. ∎

**b. Sufficiency.** On the basis of the ZGDS strategy (10), we can derive that:

$$\sum_{i=1}^{N} [\nabla g\big(W_t^i(k)\big) - \nabla g\big(W_t^i(k-1)\big)] = \sum_{i=1}^{N} \nabla g\big(W_t^i(k)\big) - \sum_{i=1}^{N} \nabla g\big(W_t^i(k-1)\big) = \mathbf{0}_n,$$

$$\vdots$$

$$\sum_{i=1}^{N} [\nabla g\big(W_t^i(2)\big) - \nabla g\big(W_t^i(1)\big)] = \sum_{i=1}^{N} \nabla g\big(W_t^i(2)\big) - \sum_{i=1}^{N} \nabla g\big(W_t^i(1)\big) = \mathbf{0}_n,$$

$$\sum_{i=1}^{N} [\nabla g\big(W_t^i(1)\big) - \nabla g\big(W_t^i(0)\big)] = \sum_{i=1}^{N} \nabla g\big(W_t^i(1)\big) - \sum_{i=1}^{N} \nabla g\big(W_t^i(0)\big) = \mathbf{0}_n. \quad (19)$$

From the recursive relationships in the formulas of (19), we can derive that:

$$\sum_{i=1}^{N} \nabla g\big(W_t^i(k)\big) - \sum_{i=1}^{N} \nabla g\big(W_t^i(0)\big)$$

$$= \sum_{i=1}^{N} \nabla g\big(W_t^i(k)\big) - \sum_{i=1}^{N} \nabla g\big(W_t^i(k-1)\big) + \cdots + \sum_{i=1}^{N} \nabla g\big(W_t^i(2)\big) - \sum_{i=1}^{N} \nabla g\big(W_t^i(1)\big)$$

$$+ \sum_{i=1}^{N} \nabla g\big(W_t^i(1)\big) - \sum_{i=1}^{N} \nabla g\big(W_t^i(0)\big)$$

$$= \mathbf{0}_n. \quad (20)$$

Based on (19) and the ZGS-based initialization (9), the ZGS strategy (8) is constantly guaranteed over the sub-iterations in an iteration step. ∎

## G    PROOF OF LEMMA 2

Consider the Stage 1 in each iteration step of the ZGS-ED-DLCB algorithm (12), the corresponding global Lyapunov function candidate is given as:

$$V_t(k) = \frac{1}{2} \sum_{i=1}^{N} \big(W_t^* - W_t^i(k)\big)^\top (S_t^{i^\top} S_t^i + \sigma I_{nd}) \times \big(W_t^* - W_t^i(k)\big). \quad (21)$$

From (21), the following inequalities can be further obtained:

$$V_t(k) \geqslant \sum_{i=1}^{N} \frac{\xi_t^i}{2} \|W_t^* - W_t^i(k)\|^2 \geqslant \frac{\xi_t}{2} \sum_{i=1}^{N} \|W_t^* - W_t^i(k)\|^2, \tag{22}$$

$$V_t(k) \leqslant \frac{\Xi_t}{2\lambda_2(\mathcal{L})} W_t(k)^\top (\mathcal{L} \otimes I_n) W_t(k). \tag{23}$$

The proof of (23) is illustrated in App. D. And the difference of $V_t(k)$ is given by

$$\triangle V_t(k+1) = V_t(k+1) - V_t(k)$$

$$= -\frac{1}{2} \sum_{i=1}^{N} \big(W_t^i(k)^\top (S_t^{i^\top} S_t^i + \sigma I_{nd}) W_t^i(k) - W_t^i(k+1)^\top (S_t^{i^\top} S_t^i + \sigma I_{nd}) W_t^i(k+1)\big). \tag{24}$$

**Remark 2.** Consider the **Stage 1** in the DL setting, we can derive that $\sum_{i=1}^{N}(S_t^{i^\top} S_t^i + \sigma I_{nd})\big(W_t^i(k+1) - W_t^i(k)\big) = \gamma_t \sum_{i=1}^{N} \sum_{j \in \mathcal{N}_i} a_{ij}\big(\hat{W}_t^j(k) - \hat{W}_t^i(k)\big) = \mathbf{0}_n.$

In order to simplify in ease, the intermediate terms are constructed in the derivation process. Therefore, we can obtain:

$$\triangle V_t(k+1) = -\frac{1}{2} \sum_{i=1}^{N} \big(W_t^i(k)^\top (S_t^{i^\top} S_t^i + \sigma I_{nd}) W_t^i(k) - W_t^i(k+1)^\top (S_t^{i^\top} S_t^i + \sigma I_{nd}) W_t^i(k+1)\big)$$

$$- 2 \sum_{i=1}^{N} \big(W_t^i(k+1) - W_t^i(k)\big)^\top (S_t^{i^\top} S_t^i + \sigma I_{nd}) W_t^i(k+1)$$

$$+ 2 \sum_{i=1}^{N} \big(W_t^i(k+1) - W_t^i(k)\big)^\top (S_t^{i^\top} S_t^i + \sigma I_{nd}) W_t^i(k+1)$$

$$= -\frac{1}{2} \sum_{i=1}^{N} \big(W_t^i(k+1) - W_t^i(k)\big)^\top (S_t^{i^\top} S_t^i + \sigma I_{nd}) \big(W_t^i(k+1) - W_t^i(k)\big)$$

$$+ \sum_{i=1}^{N} \big(W_t^i(k+1) - W_t^i(k)\big)^\top (S_t^{i^\top} S_t^i + \sigma I_{nd}) W_t^i(k+1)$$

$$\leqslant \sum_{i=1}^{N} \big(W_t^i(k+1) - W_t^i(k)\big)^\top (S_t^{i^\top} S_t^i + \sigma I_{nd}) W_t^i(k+1)$$

$$= \big(W_t(k+1) - W_t(k)\big)^\top (S_t^\top S_t + \sigma I_{Nnd}) W_t(k+1). \tag{25}$$

Combining (25) with (13), we can derive:

$$\triangle V_t(k+1) = V_t(k+1) - V_t(k)$$

$$\leqslant -\gamma_t (W_t(k) + e_t(k))^\top (\mathcal{L} \otimes I_{nd}) W_t(k+1)$$

$$= -\gamma_t W_t(k)^\top (\mathcal{L} \otimes I_{nd})[-\gamma_t (S_t^\top S_t + \sigma I_{Nnd})^{-1}(\mathcal{L} \otimes I_{nd})(W_t(k) + e_t(k)) + W_t(k)]$$

$$- \gamma_t e_t(k)^\top (\mathcal{L} \otimes I_{nd})[-\gamma_t (S_t^\top S_t + \sigma I_{Nnd})^{-1}(\mathcal{L} \otimes I_{nd}) \times (W_t(k) + e_t(k)) + W_t(k)]$$

$$= -\gamma_t W_t(k)^\top (\mathcal{L} \otimes I_{nd}) W_t(k) + \gamma_t^2 W_t(k)^\top (\mathcal{L} \otimes I_{nd})(S_t^\top S_t + \sigma I_{Nnd})^{-1}(\mathcal{L} \otimes I_{nd}) W_t(k)$$

$$+ 2\gamma_t^2 W_t(k)^\top (\mathcal{L} \otimes I_{nd})(S_t^\top S_t + \sigma I_{Nnd})^{-1}(\mathcal{L} \otimes I_{nd}) e_t(k) - \gamma_t e_t(k)^\top (\mathcal{L} \otimes I_{nd}) W_t(k)$$

$$+ \gamma_t^2 e_t(k)^\top (\mathcal{L} \otimes I_{nd})(S_t^\top S_t + \sigma I_{Nnd})^{-1}(\mathcal{L} \otimes I_{nd}) e_t(k). \tag{26}$$

Then we can derive the following two matrix inequalities from the Young's inequality Mitrinovic & Vasic (1970) (see App. J) and properties of symmetric positive definite and semidefinite matrices that

$$W_t(k)^\top (\mathcal{L} \otimes I_{nd})(S_t^\top S_t + \sigma I_{Nnd})^{-1}(\mathcal{L} \otimes I_{nd}) e_t(k)$$

$$= e_t(k)^\top (\mathcal{L} \otimes I_{nd})(S_t^\top S_t + \sigma I_{Nnd})^{-1}(\mathcal{L} \otimes I_{nd}) W_t(k)$$

$$\leqslant \frac{\lambda_{\max}(\mathcal{L})}{\xi_t} e_t(k)^\top (\mathcal{L} \otimes I_n) W_t(k), \tag{27}$$

and

$$e_t(k)^\top (\mathcal{L} \otimes I_n) W_t(k) \leqslant \frac{\lambda_{\max}(\mathcal{L})}{2\epsilon} W_t(k)^\top (\mathcal{L} \otimes I_n) W_t(k) + \frac{\epsilon}{2} e_t(k)^\top e_t(k), \tag{28}$$

where $\lambda_{\max}(\mathcal{L}) = \lambda_{\max}(\mathcal{L})$, $\xi_t = \lambda_{\min}(S_t^\top S_t + \sigma I_{Nnd})$ and $\epsilon > 0$. By substituting the inequalities (27) and (28) into the inequality (26) together with the condition $\gamma_t > \xi_t / 2\lambda_{\max}(\mathcal{L})$, we have:

$$\Delta V_t(k+1) \leqslant - \gamma_t \eta_1 W_t(k)^\top (\mathcal{L} \otimes I_{nd})^\top W_t(k) + \eta_2 e_t(k)^\top e_t(k), \tag{29}$$

in which $\eta_1 = 1 - \frac{\gamma_t \lambda_{\max}(\mathcal{L})}{\xi_t} - \frac{2\gamma_t \lambda_{\max}(\mathcal{L})^2 - \xi_t \lambda_{\max}(\mathcal{L})}{2\xi_t \epsilon} > \frac{1}{2}$ and $\eta_2 = \frac{\gamma_t}{2\xi_t}[2\gamma_t \lambda_{\max}(\mathcal{L})^2 + (2\gamma_t \lambda_{\max}(\mathcal{L}) - \xi_t)\epsilon]$.

On the basis of the trigger function (11) and the conditions $\frac{\xi_t}{2\lambda_{\max}(\mathcal{L})} < \gamma_t < \min\{\frac{\Xi_t}{2\lambda_2(\mathcal{L})}, \frac{\xi_t(2\epsilon+\lambda_{\max}(\mathcal{L}))}{2\lambda_{\max}(\mathcal{L})(\epsilon+\lambda_{\max}(\mathcal{L}))}\}$ and $\epsilon > 0$, we derive that $\eta_1 \in (\frac{1}{2}, 1)$ and $e_t(k)^\top e_t(k) \leqslant N\alpha \beta_t^{\ k}$. Then, from (23), we can derive:

$$\Delta V_t(k+1) \leqslant - \frac{2\lambda_2(\mathcal{L})\gamma_t \eta_1}{\Xi_t} V_t(k) + N\eta_2 \alpha \beta_t^{\ k}. \tag{30}$$

Further, we obtain:

$$V_t(k+1) \leqslant \theta_t V_t(k) + N\eta_2 \alpha \beta_t^{\ k}, \tag{31}$$

where $\theta_t = 1 - \frac{2\lambda_2(\mathcal{L})\gamma_t \eta_1}{\Xi_t}$.

Based on the aforementioned results, if the conditions $\frac{\xi_t}{2\lambda_{\max}(\mathcal{L})} < \gamma_t < \min\{\frac{\Xi_t}{2\lambda_2(\mathcal{L})}, \frac{\xi_t(2\epsilon+\lambda_{\max}(\mathcal{L}))}{2\lambda_{\max}(\mathcal{L})(\epsilon+\lambda_{\max}(\mathcal{L}))}\}$ and $\epsilon > 0$ are satisfied, we can derive that $\theta_t \in (1 - \eta_1, \frac{1-\lambda_2(\mathcal{L})\xi_t \eta_1}{\lambda_{\max}(\mathcal{L})\Xi_t}) \subset (0,1)$ and

$$\begin{aligned} V_t(k) &\leqslant \theta_t V_t(k-1) + N\eta_2 \alpha \beta_t^{\ k-1} \\ &\leqslant \theta_t^{\ 2} V_t(k-2) + \theta_t N\eta_2 \alpha \beta_t^{\ k-2} + N\eta_2 \alpha \beta_t^{\ k-1} \\ &\vdots \\ &\leqslant \theta_t^{\ k} V_t(0) + N\eta_2 \alpha(\theta_t^{\ k-1}\beta_t^{\ 0} + \theta_t^{\ k-2}\beta_t^{\ 1} + \ldots + \theta_t^{\ 1}\beta_t^{\ k-2} + \theta_t^{\ 0}\beta_t^{\ k-1}) \\ &= \theta_t^{\ k} V_t(0) + N\eta_2 \alpha \frac{\theta_t^{\ k} - \beta_t^{\ k}}{\theta_t - \beta_t} \\ &= (V_t(0) - \rho_t)\theta_t^{\ k} + \rho_t \beta_t^{\ k}, \end{aligned} \tag{32}$$

in which $\rho_t = \frac{N\eta_2 \alpha}{\beta_t - \theta_t}$. According to the inequality (22), the inequality (15) in Lemma 2 can be derived. Therefore, the proof is completed. ∎

## H  PROOF OF THE INEQUALITY (23)

It can be derived from Conclusion 2 (see App. K) that:

$$\begin{aligned} \sum_{i=1}^{N} \|W_t^i(k) - \frac{1}{N}\sum_{j=1}^{N} W_t^j(k)\|^2 &= \frac{1}{N}[W_t^1(k)^\top, W_t^2(k)^\top, \cdots, W_t^N(k)^\top](\bar{\mathcal{L}} \otimes I_n) \\ &\qquad \times [W_t^1(k)^\top, W_t^2(k)^\top, \cdots, W_t^N(k)^\top]^\top \\ &= \frac{1}{N} W_t(k)^\top (\bar{\mathcal{L}} \otimes I_n) W_t(k) \\ &\leqslant \frac{1}{\lambda_2(\mathcal{L})} W_t(k)^\top (\mathcal{L} \otimes I_n) W_t(k). \end{aligned} \tag{33}$$

Combine (28) with Lemma 4 (see App. L), we obtain:

$$V_t(k) \leqslant \frac{\Xi_t}{2\lambda_2(\mathcal{L})} W_t(k)^\top (\mathcal{L} \otimes I_n) W_t(k). \tag{34}$$

∎

## I   PROOF OF LEMMA 3

Let $x$ be an eigenvector of $S_t^\top S_t$ corresponding to an eigenvalue $\lambda$, and we have that $(S_t^\top S_t)x = \lambda x$. Then $(S_t^\top S_t + \sigma I_{Nnd})x = (S_t^\top S_t)x + (\sigma I_{Nnd})x = \lambda x + \sigma I_{Nnd}x = \lambda x + \sigma x = (\lambda + \sigma)x$. Thus, $\lambda + \sigma$ is an eigenvalue of $S_t^\top S_t + \sigma I_{Nnd}$. As $S_t^\top S_t$ is an $Nnd \times Nnd$ nonnegative matrix, its $Nnd$ eigenvalues are nonnegative, i.e., $\lambda \geqslant 0$. Therefore, $\sigma$ is the infimum (or the greatest lower bound) of the eigenvalues of $S_t^\top S_t + \sigma I_{Nnd}$. As $\xi_t = \min_{i \in \mathcal{V}} \xi_t^i = \lambda_{\min}(S_t^\top S_t + \sigma I_{Nnd})$ and $\xi_t^i = \lambda_{\min}(S_t^{i\top} S_t^i + \sigma I_{nd})$ in Lemma 2, $\sigma$ is also the infimum of $\xi_t$ and $\xi_t^i$. ∎

## J   YOUNG'S INEQUALITY MITRINOVIC & VASIC (1970)

If real numbers $a, b \geqslant 0$ and if real numbers $p, q \geqslant 1$ and satisfy $\frac{1}{p} + \frac{1}{q} = 1$, then

$$ab \leqslant \frac{a^p}{p} + \frac{b^q}{q}. \tag{35}$$

## K   USEFUL CONCLUSIONS

**Conclusion 1: Equivalent statements and implications for strong convexity of functions** Lu & Tang (2012); Chen & Ren (2016); Zhou (2018).

**a. Equivalent statements:**
(1) $f : \mathbb{R}^n \to \mathbb{R}$ is strongly convex with a parameter $\mu$;
(2) $f(y) \geqslant f(x) + \nabla f(x)^\top (y - x) + \frac{\mu}{2}\|y - x\|^2$ for $\forall x, y \in \mathbb{R}^n$;
(3) $\left(\nabla f(y) - \nabla f(x)\right)^\top (y - x) \geqslant \mu\|y - x\|^2$ for $\forall x, y \in \mathbb{R}^n$.   □

**b. Equivalent implications:**
(1) $f$ is strongly convex with a parameter $\mu$ and continuously differentiable;
(2) $f(y) \geqslant f(x) + \nabla f(x)^\top (y - x) + \frac{\mu}{2}\|y - x\|^2$ for $\forall x, y \in \mathbb{R}^n$, and has a minimum $x^*$;
(3) $\left(\nabla f(y) - \nabla f(x)\right)^\top (y - x) \geqslant \mu\|y - x\|^2$ for $\forall x, y \in \mathbb{R}^n$, and has a minimum $x^*$.   □

**Conclusion 2** Lu & Tang (2012). For any graph $\mathcal{G} \in \mathbb{G}$ with $N$ vertices and its complete graph $\bar{\mathcal{G}}$, $\mathcal{L}$ has $N - 1$ positive eigenvalues and there exists a $W \in \mathbb{R}^{N \times N}$, which includes $N$ orthonormal eigenvalues of $\mathcal{L}$ in its columns. Then $W^\top \bar{\mathcal{L}} W$ and $W^\top \mathcal{L} W$ are diagonal matrices similar to $\mathcal{L}$ and $\bar{\mathcal{L}}$, and the eigenvalue 0 is located in the same position, respectively. Therefore, $\lambda_2(\mathcal{L})W^\top \bar{\mathcal{L}} W \leqslant N W^\top \mathcal{L} W$. "$\leqslant$" denotes the relationship between diagonal elements at the same locations.   □

## L   LEMMAS

**Lemma 4.** For the undirected connected MAS topology, when $\sum_{i=1}^N \nabla g\left(W_t^i(k)\right) = \mathbf{0}_n$ and $V_t(k) = \sum_{i=1}^N \left[g(W_t^*) - g\left(W_t^i(k)\right) - \nabla g\left(W_t^i(k)\right)^\top \left(W_t^* - W_i(k)\right)\right]$, the following inequality can be derived:

$$V_t(k) \leqslant \sum_{i=1}^N \frac{\Xi_t^i}{2}\|W_t^i(k) - \frac{1}{N}\sum_{j=1}^N W_t^j(k)\|^2 \leqslant \frac{\Xi_t}{2}\sum_{i=1}^N \|W_t^i(k) - \frac{1}{N}\sum_{j=1}^N W_t^j(k)\|^2. \tag{36}$$

□

## M   REGRET BOUND FOR ZGS-ED-DLCB

Regret bound of the ZGS-ED-DLCB algorithm originates from to the theory of Regret Bound in Section 4.2 of Bogunovic et al. (2016). In terms of the squared exponential (SE) kernel, the regret

bound of the ZGS-ED-DLCB algorithm is $R_T = \tilde{\mathcal{O}}(\max\{\sqrt{T}, T\epsilon^{1/6}\})$ when employing the local LCB acquisition functions and RFF-based surrogate model according to Corollary 4.1 of Bogunovic et al. (2016).

## N  CONSENSUS VS. AVERAGE CONSENSUS

In the area of *distributed learning* and distributed optimziation, many works are devoted to the problems of consensusYu & Chen (2020); Nedić et al. (2018); Ren et al. (2018); Chen & Ren (2016); Lu et al. (2011); Nedic et al. (2010); Olfati-Saber et al. (2007) and average consensus Koloskova et al. (2019a;b); Scardapane et al. (2015); Aysal et al. (2008); Xiao et al. (2007); Xiao & Boyd (2004) between agents, respectively. There exists essential difference between consensus and average consensus.

In terms of $W_t^i(k)$ in this work, consensus in each iteration step means that $\lim_{k\to\infty} W_t^i(k) = W_t^{i*}$ (practically, $k = K \in \mathbb{Z}_+$), but average consensus in each iteration step means that $\lim_{k\to\infty} W_t^i(k) = \frac{1}{N}\sum_{i=1}^N W_t^i(0)$. Thus, average consensus derives the average of local optimum (or local initial values), but not the global optimum, i.e., consensus in theory.

Typically, average consensus-based algorithms have some applications in practice due to their fast convergence rate (e.g., exponential convergence). Whereas consensus-based algorithms with much more convergence accuracy involves more theoretical designs and analysis on convergence robustness and convergence rate.

## O  BOCHNER'S THEOREM RASMUSSEN ET AL. (2006)

In terms of a weakly stationary mean square continuous complex-valued random process on $\mathbb{R}^D$, a complex-valued function $k(\boldsymbol{\tau})$, $\boldsymbol{\tau} \in \mathbb{R}^D$, is the covariance function of the random process iff $k(\boldsymbol{\tau})$ can be expressed as:

$$k(\boldsymbol{\tau}) = \int_{\mathbb{R}^D} e^{2\pi i \boldsymbol{s} \cdot \boldsymbol{\tau}} d\mu(\boldsymbol{s}) \tag{37}$$

in which $\mu(\boldsymbol{s})$ is a positive finite measure.

## P  EXPERIMENTS

### P.1  EXPERIMENTAL DETAILS

**Experimental Environment.** All the experiments are implemented by the Pycharm Software on Windows OS with 16GB RAM and the results can be reproduced by our code and data that involved in the material. More details are shown in the code.

**Modeling.** ZGS-ED-DLCB is an online learning algorithm in theory with a complete set of time-varying parameters except $a_{ij}$ (see (12)). As the baseline algorithms are offline learning with high certainty, we extend the offline baselines to the corresponding online learning (i.e, online hyperparameter tuning) algorithms for comparing with ZGS-ED-DLCB. Thus, the learning performance of baselines are enhanced than their original versions in theory. For further comparing in justice, the baselines also adopt the RFF-based surrogate model because the RFF-based model can also deal with traditional learning with high certainty. In terms of the synthetic experiments, the observation outputs of the benchmark functions are assumed to be noise-free.

### P.2  SUPPLEMENTARY EXPERIMENTS

Detailed experiments illustrating the average triggering frequency through the iterations for all trials and the triggering instants at randomly selected iteration steps in each trial are depicted in Fig. 1-3. Thus, the communication efficiency of the event-driven mechanism is further verified.

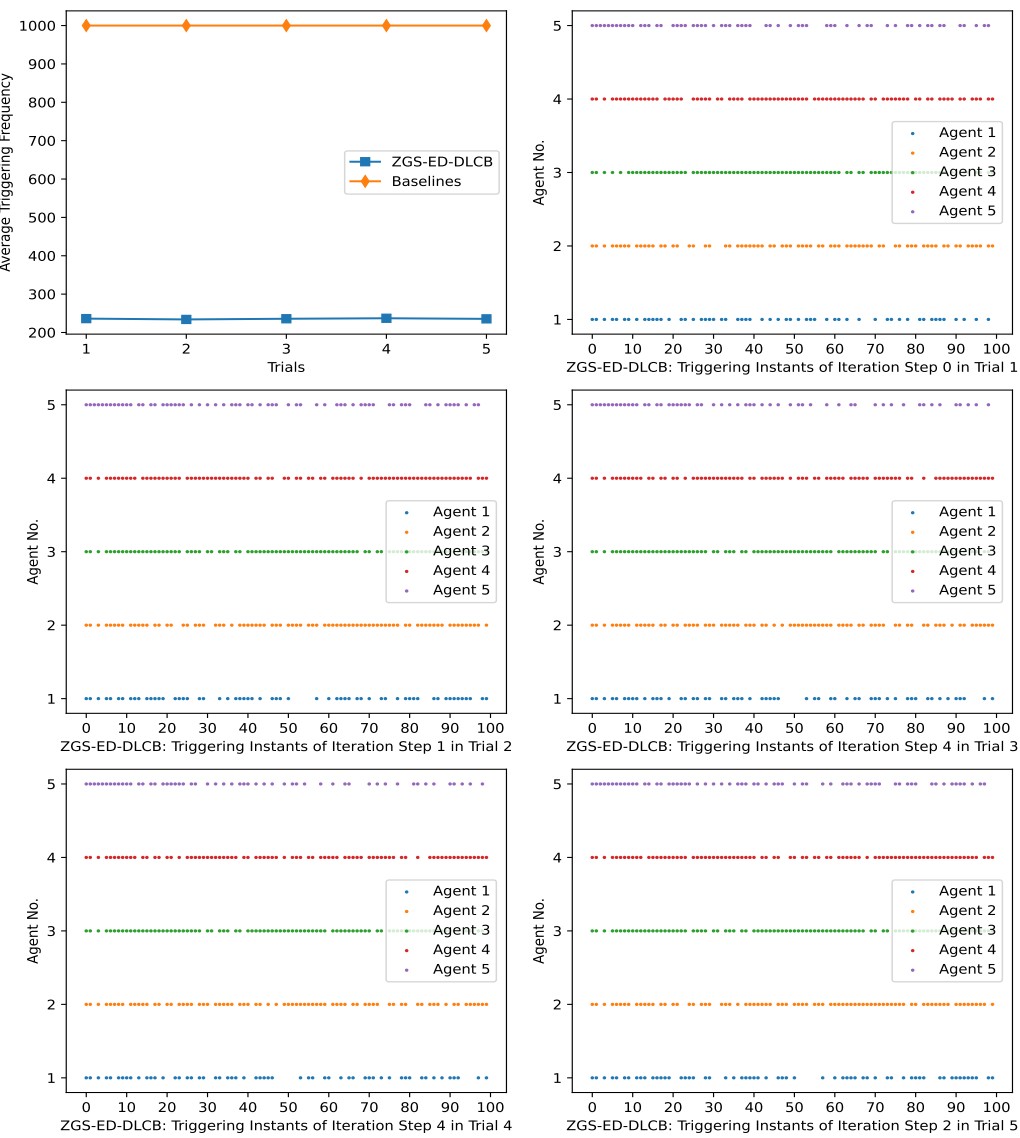

Figure 1: Average triggering frequency for the 5 trials & Exhibition of triggering instants at the randomly selected iteration steps wrt the Levy function in the complete topology.

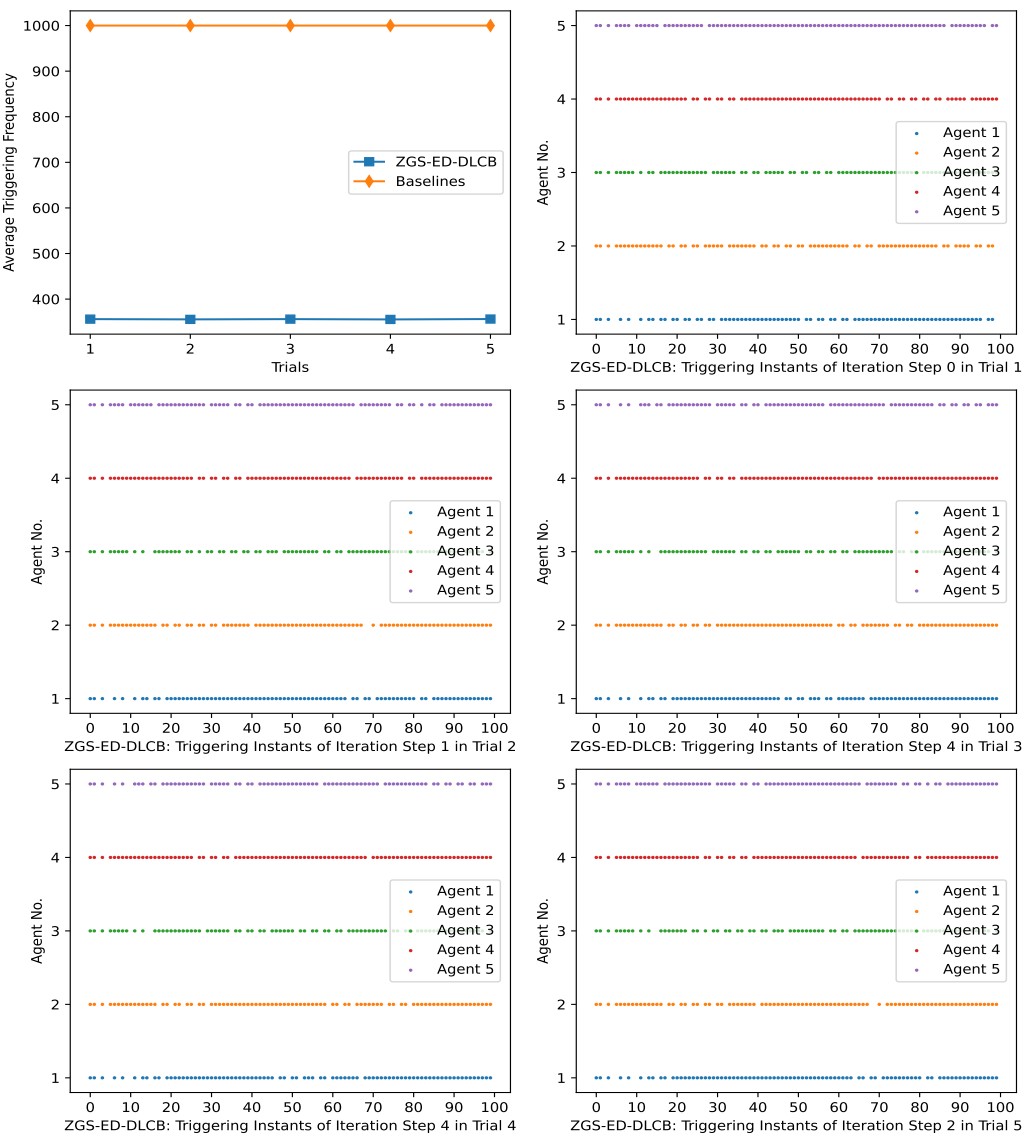

Figure 2: Average triggering frequency for the 5 trials & Exhibition of triggering instants at the randomly selected iteration steps wrt the Ackley function in the complete topology.

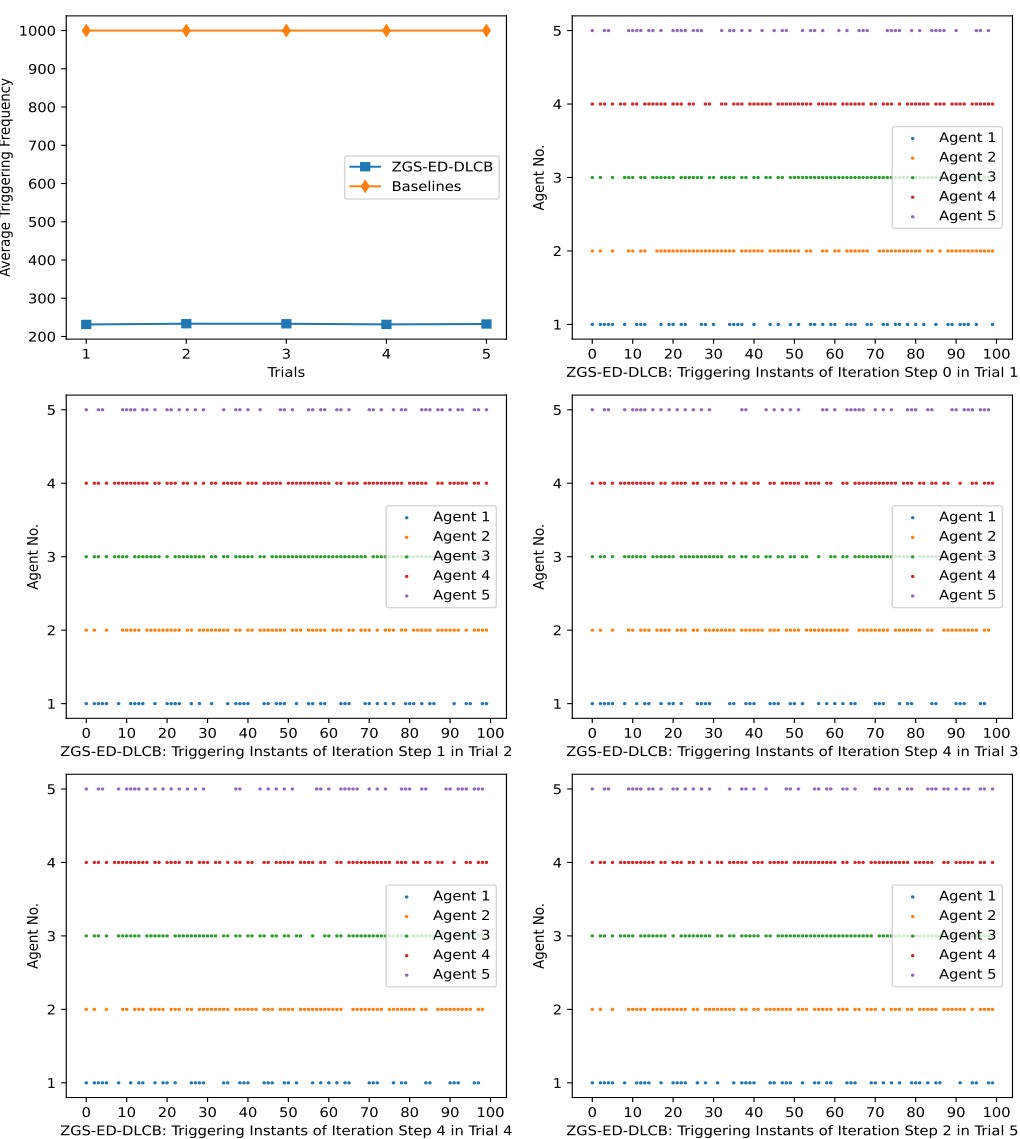

Figure 3: Average triggering frequency for the 5 trials & Exhibition of triggering instants at the randomly selected iteration steps wrt the Griewank function in the complete topology.

