# OpenReview forum: "ZGS-Based Event-Driven Algorithms for Bayesian Optimization in Fully Distributed Multi-Agent Systems"
_ICLR.cc/2024/Conference — Submitted to ICLR 2024_

### Official Review · Reviewer_BqAx · 2023-10-21

**Soundness:** 1 poor
**Presentation:** 1 poor
**Contribution:** 2 fair
**Rating:** 1
**Confidence:** 3

**Summary:**

The authors study the problem of distributed BO, in which multiple agents wish to optimize the same black-box objective function in a sample-efficient manner but cannot share their data due to privacy concerns. The authors propose ZGS-ED-DLCB, an algorithm that circumvents the problem by sharing the parameters of a random Fourier features model instead of the data directly. The algorithm builds off an event-triggered zero-gradient sum algorithm in the distributed learning literature. The authors provide theoretical results to show the convergence of the parameters of all agents in each iteration. The authors compare ZGS-ED-DLCB to previous algorithms in the distributed learning literature.

**Strengths:**

1. Applying the event-triggered zero-gradient-sum algorithm to BO may be of interest to BO practitioners interested in distributed BO with privacy concerns.
2. The authors show that their algorithm outperforms previous algorithms in the distributed learning literature, suggesting that their algorithm has empirical advantages within the BO setting as compared to these other algorithms.

**Weaknesses:**

The paper has several major weaknesses:

1. **Technical misconception of random Fourier features (RFF) in a BO context.** Section 3.3, Stage 2 says "Based on the obtained $W_t^{i*}$, each agent calculates the aforementioned mean and prediction variance at the iteration step t. Using the calculated $m_t(\mathbf x^*)$ and $\hat \sigma_t^2(\mathbf x^*)$, the RFF-based DLCB acquisition function is employed..." where DLCB is a UCB-type acquisition (lower bound version). $W_t^{i*}$ are the parameters to the linear model that arises from a RFF approximation of the GP ($\omega$ in [1]); specifically, the parameters that minimize the linear model's regularized L2 loss with respect to all agents' data. The problem arises as $W_t^{i*}$ has nothing to do with the computation of $m_t(\mathbf x)$ and $\hat \sigma_t^2(\mathbf x)$. In a RFF context, $m_t(\mathbf x)$ and $\hat \sigma_t^2(\mathbf x)$ are computed only using $s$ ($\phi$ in [1]), i.e., the RFF approximation to the kernel $k$, along with the data (see Appendix B in [1]). This begs the question, what is the use of computing $W_t^{i*}$ if a UCB algorithm is to be used subsequently? Computing $W_t^{i*}$ makes sense in a Thompson sampling context (as was done in [1]), but not with UCB algorithms. This renders the significance of most of the technical content in this paper questionable since most of the paper is devoted to the ZGS-based algorithm for computing $W_t^{i*}$.

2. **Experiments do not compare to the most relevant baselines, and results are not presented well.** The paper makes mention of [1] and [2], and the idea of using RFF to share information in a parametric manner is borrowed from [1]. In a BO context, [1] is clearly the closest work. However, the experiments do not compare against the algorithms from [1] and [2], and instead compare against existing algorithms in the distributed learning (DL) literature. This is a problematic choice given that, in the introduction, the authors claim that "the existing DL works ... do not consider the expensive black-box optimization problems utilizing limited data", so why only use them for the experiments and exclude the directly relevant algorithms from [1] and [2]? This weakens the empirical support for the proposed algorithm. Furthermore, the authors present the number $R_T/T$ without error bars in tables instead of graphs of the per iteration cumulative regret as is standard for BO works in order to compare the convergence over time.

3. **Significance of theory is questionable.** From a BO perspective, this paper does not contribute anything theory-wise. All the results are about the convergence of the parameters $W$ in each iteration due to the ZGS algorithm, and it is not clear how this suggests sublinear regret. There is a cryptic paragraph about a regret bound in Appendix M (not referenced from the main paper) which handwaves the issue to the results from another paper. From a distributed learning perspective, I am not sure that this paper has significant contribution, since the BO problem is turned into a linear regression problem upon which the ZGS algorithm is applied, but the ZGS algorithm builds upon work in [3] which is for continuously differentiable strongly convex problems, a more general class of problems.

4. **Severe clarity issues**:

    a. Problems with notation. Some variables are used before they are defined e.g. $W(k)$, and some are never defined at all e.g. $\hat \sigma_2(\mathbf x)$. Some notation is inconsistent e.g. $\mathbf x$ and $x$ are both for inputs; vectors are sometimes denoted in lower case, sometimes they are in bold lower case, sometimes they are in upper case. Some notation is just wrong, e.g., in Appendix B, an input output pair is written as $(\mathbf x_1, f(\mathbf x_1))$, and then the output is later referred to as $y_1$, but $f(\mathbf x_1)$ and $y_1$ are separate quantities, the latter is the noise-corrupted version of the former.

    b. Extensive typos and grammatical errors, including but not limited to: "addressing these significant gaps, we aim to design a DBO algorithm that solves costly-to-evaluate black-box optimization problems in a fully distributed multi-agent system (MAS) is nontrivial and promising"; "It is the first time to provide"; "2.1 ALTERNATIVE TO A GAUSSIAN PORCESS"; "As aforementioned discussions".


[1] Dai et. al., 2020. "Federated Bayesian optimization via Thompson sampling".

[2] Dai et. al., 2020. "Differentially private federated Bayesian optimization with distributed exploration".

[3] Chen and Ren, 2016. "Event-triggered zero-gradient-sum distributed consensus optimization over directed networks".

**Questions:**

No questions other than those listed in the Weaknesses section.

---

> ### Author Response · Authors · 2023-11-21
>
> (1). Bayesian optimization (BO) aims to optimize the function indeed, and we design the novel distirbuted BO (DBO) paradigm in order to optimize the global objective function (7) of DBO (equally $f$ in (4)) over a fully distributed multi-agent system (MAS). The whole procedure of BO includes surrogate modeing and derserving next potential profitable locations by acquisition functions. Gaussian process (GP) is the mainstream choice for surrogate modeling. Acquisition functions are utilized to search next potential profitable locations. In our work, Random Fourier features (RFF) are used to replace a typical GP of BO. The reference paper "Dai et al., 2020" illustrates that "A GP with RFF approximation can be interpreted as a Bayesian linear regression model with $\phi(x)$ as the features". However, unlike that W are sampled according to (2) of "Dai et al., 2020" in the FBO setting, $W$ is derived by training in our DBO setting, specifically, training by our presented ZGS-ED-DLCB according to (12). Then the local RFF-based DLCB is utilized to deserve the next profitable location by using the trained $\hat{W}_{t}^{i}(K)$ for each agent in each iteration step.
>
> (2). Thanks a lot for reading the reference paper "Dai et al. (2020)", which is [1] in your mark. The reference paper [2] you mark is "Dai et al. (2021)" in our reference list. Although both tree-structured federated learning (FL) and decentralized (fully) distributed learning (DL) are distinct distributed machine learning paradigms, FBO in FL and DBO in DL are distinct paradigms. FBO/FL algorithms cannot be implemented in the DBO/DL setting. We reference Dai et al. (2021; 2020), however, it's not feasible to conduct comparative experiments using the algorithms presented in those references. We introduce and investigate the novel DBO paradigm, and we present the first DBO algorithm, ZGS-ED-DLCB. No prior DBO algorithms have been developed before. Similar to the comparative experiments in Dai et al. (2020), well-known DL algorithms including the DCL, ATC-LMS, CTA-LMS and ADMM-based algorithms are adjusted to adapt to the DBO setting as the baselines in the comparative experiments.
>
> Moreover, in the FBO experiments, none of the baselines compared in the experiments are federated/distributed BO. The following is a direct quote from the NeurIPS reference paper Dai et al. (2020): "Although FTS is the first algorithm for the FBO setting, some algorithms for meta-learning in BO, such as ranking-weighted GP ensemble (RGPE) [16] and transfer acquisition function (TAF) [55], can be adapted to the FBO setting through a heuristic combination with RFF approximation.". Drawing inspiration from the experimental design in Dai et al. (2020), we have constructed comparative experiments for the first DBO algorithm.
>
> (3). We consider the expensive-to-evaluate black-box optimization problem within a fully distributed multi-agent system (MAS) and create a distributed BO (DBO) algorithm, ZGS-ED-DLCB. Moreover, no prior works inverstigate the problem (including leveraging BO) in a fully distributed MAS. In very recent years, novel federated Bayesian optimization (FBO) algorithms, which precede the development of DBO algorithms, (Dai et al. (2021; 2020)) are developed in the federated learning (FL) setting. Tree-structured federated learning (FL) and decentralized (fully) distributed learning (DL) are distinct distributed machine learning paradigms. Similiar to that Dai et al. (2020) presents the FBO paradigm and the first FBO algorithm in the FL setting, we propose the DBO paradigm and the first DBO algorithm in the DL setting. In contrast to the FBO in the FL setting, DBO faces unique challenges to tackle, which are illustrated in the Introduction section. At the end of Introduction section, the main contributions are summerized.
>
> (4). In response to your concerns, we have revisited our manuscript with a fresh perspective and revised it. If you are interested, the final version will deliever the revised manuscript.

---

> > ### Comment · Reviewer_BqAx · 2023-11-22
> >
> > Thanks for your response.
> >
> > 1) My question hasn’t been answered. The algorithm from Dai et. al. (2020) is Thompson sampling where $W^i_t{}^*$ is used. Your algorithm is a UCB variant where $W^i_t{}^*$ is not used and only $m_t$ and $\hat{\sigma}^2$ are used. Thompson sampling and UCB are different algorithms. $W^i_t{}^*$ is not involved in the computation of $m_t$ and $\hat{\sigma}^2$. Why exactly is $W^i_t{}^*$ computed in your algorithm at every iteration?
> >
> > 2) Why exactly is the FBO setting incompatible with the DBO setting? Please describe the incompatibility precisely by comparing their problem formulations. Surely the algorithm from Dai et. al. (2020) can be adapted to the DBO setting, even if assuming a specific network topology is necessary.
> >
> > 3) You may be the first to combine the DL setting with BO, but this does not address the issue that the theoretical results in Section 4 are not significant from either a BO or a DL perspective.

---

> ### Author Response · Authors · 2023-11-23
>
> Thanks for your interests and comments.
>
> Response:
>
> 1. ${W_t^i}^*$ is derived by training and used in each iteration step of ZGS-ED-DLCB. Specifically, it is utilized in the calculation of  local mean function $m_{i, t}\left(\mathrm{x}\right)=S_t^i {W_t^i}^*$ (please refer to the equation (5) in our work) for each agent. The calculation of $m_{i, t}\left(\mathrm{x}\right)$ is eaisier and more resource-saving than that of the FBO algorithms. In terms of the aforementioned corresponding or relevant content, we have revisited our manuscript with a fresh perspective and revised it. If you are interested, the final version will deliever the revised manuscript.
>
> 2. The problem formlation can be the same for FBO and DBO. However, while FBO uses a tree-structured paradigm with a cloud server, DBO is a decentralized paradigm without the use of a server. The FBO setting includes a cloud server, while no server is leveraged in the DBO setting. The procedure of an FBO algorithm includes the agent side and the server side, whereas the procedure of a DBO algorithm only involves the agent side. The distinction extends beyond just the network topology. There are specific communication mechanisms and limitations in the settings of fully distributed learning (DL) and federated learning (FL), not just in the aspect of network topology. Therefore, FBO algorithms are not incompatible with the DBO setting, and vice versa. FL origins much later than DL, although they are both distributed machine learning paradigms. According to your concern, why are the FBO algorithms not compared with the adapted DL algorithms in the FBO works? It seems that comparative experiments involving the adapted well-known DL algorithms (should be adapted as you expect) in the FL setting should be implemented. But the authors of the FBO works do not handle these and it is not unreasonable due to the distinction of the DBO and FBO settings.
>
> 3. Thank you for noticing the phrase "the first". We highlight the sentence in our abstract "Moreover, we propose a novel generalized fully distributed convergence theorem, which represents a substantial theoretical and practical breakthrough wrt the ZGS-based DL." to emphasize that our ZGS-ED-DLCB algorithm overcomes the drawback exists in the recent ZGS-based DL algorithms that we notice theoretically and practically, in which all agents require the $S_{t}^{i}$ from the other agents to work. ZGS-ED-DLCB is a feasible algorithm in practice.

---

### Official Review · Reviewer_EeJ5 · 2023-10-30

**Soundness:** 3 good
**Presentation:** 2 fair
**Contribution:** 3 good
**Rating:** 6
**Confidence:** 2

**Summary:**

***** I do not have enough background knowledge to provide a fair review of this paper, since it is totally out of my scope. I sincerely request the AC to introduce another reviewer. I have set my confidence score to be 2 to indicate this.*******


This paper studies Bayesian optimization method for the fully distributed multi-agent systems. A new algorithm is proposed and the convergence is proved.

**Strengths:**

The problem setting is quite clear, and the authors describe their method straightforwardly.

The theoretical results are provided, an upper bound of the regret is given.

The experiment part is strong and shows improved results.

**Weaknesses:**

I am quite confuse about the comparison of the results to other papers. For example, is the regret bound in (15) matching the SOTA? I think some discussion on the related works and results are missing.

**Questions:**

See above.

---

> ### Author Response · Authors · 2023-11-21
>
> Response for Strengths:
>
> We greatly appreciate your positive remarks on the problem setting, theoretical results and experimental aspects of our work.
>
> Response for Weaknesses/Questions:
>
> In contrast to centralized BO, DBO is the corresponding fully distributed version. In the theory of distributed learning, learning performance of distributed learning equals to that of centralized learning. DBO does not change the regret bound of the centralized BO. Regret bound of the ZGS-ED-DLCB algorithm originates from to the theory of Regret Bound in Section 4.2 of Bogunovic et al. (2016). In terms of the squared exponential (SE) kernel, the regret bound of the ZGS-ED-DLCB algorithm is $R_T=\tilde{\mathcal{O}}(\max\lbrace\sqrt{T}, T\epsilon^{1/6}\rbrace)$ when employing the local LCB acquisition functions and the RFF-based surrogate model according to Corollary 4.1 of Bogunovic et al. (2016). Please see App. M.

---

### Official Review · Reviewer_ugDi · 2023-10-31

**Soundness:** 2 fair
**Presentation:** 1 poor
**Contribution:** 2 fair
**Rating:** 3
**Confidence:** 4

**Summary:**

This paper studies distributed Bayesian optimization in a multi-agent setting where private raw data cannot be shared among neighboring agents. The authors introduce the ZGS-ED-DLCB algorithm to address the challenge of expensive black-box optimization over a network. In this context, agents possess local data and must collaborate to achieve a global objective. The algorithm employs an event-driven mechanism to trigger communication between agents when necessary, thereby reducing the overall communication burden within the network. The paper includes theoretical and experimental results to demonstrate the efficiency and effectiveness of the proposed algorithm.

**Strengths:**

The problem-setting appears interesting and worthy of exploration. The authors provide some theoretical analysis for their algorithm, ZGS-ED-DLCB.

**Weaknesses:**

- I have to say the paper is not well written. It suffers from a lack of clarity in terms of writing, notation, and explanation, making it challenging to follow. From the problem formulation to the theoretical results, several essential details are omitted. Understanding the context is difficult, necessitating multiple visits to the appendices. I would suggest the authors maintain the core message within the main paper while relocating unnecessary sections to the appendix.

- Furthermore, the main technical contributions remain poorly explained, with insufficient references to existing literature. High-level intuitions are well presented. It is hard for me to evaluate the theoretical contribution.

- Regarding the contributions you claim, it is unclear how your algorithm tracks expensive-to-evaluate black-box optimization. Additionally, the paper does not adequately demonstrate how your algorithm handles limited data and how it compares to existing approaches. The privacy preservation aspects are mentioned, but the extent and specific mechanisms are not detailed.

Some minor comments:
- The line immediately following equation (11) introduces $\hat{W}_t^i$, but this term is not defined.
- The introduction of the notion of "regret" raises questions as the paper does not present the associated results.

**Questions:**

See weaknesses

---

> ### Author Response · Authors · 2023-11-21
>
> Response for Strengths:
>
> We appreciate your positive remarks about the problem setting and theoretical analysis of the presented ZGS-ED-DLCB algorithm in our work.
>
> Response for Weaknesses/Questions:
>
> (1) We appreciate your feedback regarding our writing and presentation. We have revisited our manuscript with a fresh perspective and revised it. If you are interested, the final version will deliever the revised manuscript.
>
> (2) We have revisited our manuscript with a fresh perspective and revised it. DBO is a novel paradigm that we introduce and investigate, no prior DBO algorithms or exist. Could you please display the additional reference papers you think that we should reference to us?
>
> (3) In our paper, the data samples used in the whole experiments are limited to each agent. Compared with the traditional centralized BO, the decentralized learning framework in the DBO setting has the following advantages in terms of privacy:
>   (a) Data Localization: In fully distributed learning, each node processes its own data locally and shares only model updates (like weight updates or gradients), not the raw data. This means that raw data doesn't need to propagate through the network, reducing the risk of data leaks.
>   (b) Reduced Centralization Risk: In centralized learning, all data is brought to a central node for processing. If this central node is compromised, all data could potentially be exposed. In fully distributed learning, there's no such central node, reducing the overall attack surface of the system.
>   (c) Regulatory Compliance: In some cases, the storage and processing of data may be subject to strict regulatory requirements, such as GDPR. In these cases, distributed learning allows for data to be processed locally, making it easier to meet these regulatory requirements.
>
> Response for Minor Comments:
>
> (1) The latest transferred knowledge of agent $i$ is denoted as $\hat{{W_t^i}}(k)=W_{t}^{i}(k_{t,i}^{m_{i}})$. We have revisited our manuscript with a fresh perspective and revised it.
>
> (2) We are the first to present nontions of regret in the DBO setting. In the synthetic experiments, the evaluation metric $R_T/T$, which is associated with the global cumulative regret $R_T$, is used to compare the learning performance of ZGS-ED-DLCB with the baselines is in the paper.

---

### Official Review · Reviewer_N4Gg · 2023-11-01

**Soundness:** 2 fair
**Presentation:** 1 poor
**Contribution:** 2 fair
**Rating:** 3
**Confidence:** 4

**Summary:**

This paper introduces a distributed optimization algorithm for multi-agent systems, in which the agents use random Fourier features to model the objective function as a linear function in the random features, and collaboratively estimate the parameter for the linear function.

**Strengths:**

- The paper derives theoretical guarantees for the parameter estimation.
- The experimental results do achieve improvements over previous methods.

**Weaknesses:**

- The most important concern I have is regarding the writing and presentation of the paper. I'm relatively familiar with the field of Bayesian optimization, but I find the paper not easy to understand and I am somewhat confused as to what is the connection between the proposed method and Bayesian optimization. If I understand correctly, the paper models the objective function $f$ as a linear function in the random features (equation 4) with parameters $W$, and the proposed algorithm aims to estimate the parameters $W$ (using the methods from equations 12 and 13) and the theoretical guarantees in Section 4 also guarantee the estimation quality for the parameters $W$. However, Bayesian optimization aims to optimize (instead of estimate) the function $f$. Therefore, it's unclear to me why the proposed method is named distributed Bayesian optimization. Also, I'm also confused by equation (7), i.e., why do we need to extend W to be time-varying? Also, none of the baselines compared in the experiments are federated/distributed BO.
- Below equation (1), the regret bound shown here is for time-varying Bayesian optimization, I wonder why is that the case?
- Section 3.2, I think the presentation of this section can be further polished. Below equation (11), what is $\hat{W}$? The estimated $W$ since the last information exchange? In the next paragraph, what is a "sub-iteration"? It is never mentioned that the proposed method has an inner loop with sub-iterations. Remark 2: again it's confusing to me why the parameters are time-varying.
- Section 3.3: I find this section not easy to understand. Perhaps it will help understanding if more intuitions/descriptions are given in addition to the equations.
- Stage 2 in Section 3.3: it says here that based on the $W^{i^*}_t$, each agent can calculate the Gaussian process posterior mean and variance. But in my understanding, with a single parameter vector $W$, we can only calculate a single function value at an input (see equation 4). To be able to calculate the GP posterior mean and variance, we need the entire distribution of $W$ (see Dai et al., (2020)). Please give the exact equations using which the GP posterior mean and variance are calculated.
- Section 4.1: I also find this section not easy to understand.

**Questions:**

Please see the questions I listed above under Weaknesses.

---

> ### Author Response · Authors · 2023-11-21
>
> Response for Strengths:
>
> We are grateful for your acknowledgement and positive assessment of the theoretical guarantees and the experimental results in our research.
>
> Response for Weaknesses/Questions:
>
> (1). We appreciate your feedback regarding our writing and presentation. Bayesian optimization (BO) aims to optimize the objective function indeed, and we design the novel distirbuted BO (DBO) paradigm in order to optimize the global objective function (7) of DBO (equally $f$ in (4)) over a fully distributed multi-agent system (MAS). The whole procedure of BO includes surrogate modeing and derserving next potential profitable locations by acquisition functions. Gaussian process (GP) is the mainstream choice for surrogate modeling. Acquisition functions are utilized to search next potential profitable locations. In our work, Random Fourier features (RFF) are used to replace a typical GP of BO. The reference paper "Dai et al., 2020" illustrates that "A GP with RFF approximation can be interpreted as a Bayesian linear regression model with $\phi(x)$ as the features". However, unlike that $W$ are sampled according to (2) of "Dai et al., 2020" in the FBO setting, $W$ is derived by training in our DBO setting, specifically, training by our presented ZGS-ED-DLCB according to (12). We call our presented paradigm DBO because we are the first to extend BO to the distributed learning (DL) setting of a fully distributed MAS, similar to that FBO origins by extending BO to the federated learning (FL) setting (please see the rederence paper "Dai et al., 2020", for example, the Abstract section).
>
> Naturally and necessarily, $W$ is time-varying in the DBO setting. $t$ represents the iteration index of ZGS-ED-DLCB and local BO. After one iteration, one potential location and its corresponding output is derived by each agent. Then, the corresponding $S_t^i$ is changed because the available data samples are increased to update $S_t^i$. Then $W$ are also necessarily changed by the next iteration step of ZGS-ED-DLCB. Thus, $W$ should be time-varying.
>
> We present the novel DBO paradigm, and we present the first DBO algorithm, ZGS-ED-DLCB. No prior DBO algorithms have been developed before. Both DBO and FBO are paradigms in the field of distributed machine learning. FL is tree-structured and DL is decentralized, so FBO algorithms cannot be implemented in the DBO setting. Similar to the reference paper "Dai et al., 2020", well-known DL algorithms including the DCL, ATC-LMS, CTA-LMS and ADMM-based algorithms are adjusted to adapt to the DBO setting as the baselines in the comparative experiments. Moreover, in the FBO experiments, none of the baselines compared in the experiments are federated/distributed BO.
>
> (2). In contrast to centralized BO, DBO is the corresponding fully distributed version. In the theory of distributed learning, learning performance of distributed learning equals to that of centralized learning. DBO does not change the regret bound of the centralized BO. Regret bound of the ZGS-ED-DLCB algorithm originates from to the theory of Regret Bound in Section 4.2 of Bogunovic et al. (2016). In terms of the squared exponential (SE) kernel, the regret bound of the ZGS-ED-DLCB algorithm is $R_T=\tilde{\mathcal{O}}(\max\lbrace\sqrt{T}, T\epsilon^{1/6}\rbrace)$ when employing the local LCB acquisition functions and the RFF-based surrogate model according to Corollary 4.1 of Bogunovic et al. (2016). Please see App. M.
>
> (3). $\hat{{W_t^i}}(k)$ is the local transferred weight vector at the $k$th sub-iteration step during the $t$th iteration step. To illustrate our ZGS-ED-DLCB algorithm more detailedly, the pseudocode is shown in App. C. From the pseudocode, we can see the the update of $W_{t}^{i}(k)$ includes $T$ iterations from 1 to $t$, of which each includes $K$ sub-iterations from 0 to $K$ in Stage 1. Thus, $t$ indexes the iteration, $k$ indexes the sub-iteration. The parameters are time-varying because lcoal data samples are actually increased after each iteration step indexed by $t$ and the parameters corresponding to the calculation process of ZGS-ED-DLCB in each iteration step are changed with the index $t$.
>
> (4). In response to your concerns, we have revisited our manuscript with a fresh perspective and revised it. If you are interested, the final version will deliever the revised manuscript.
>
> (5). RFF are utilized to approximate a typical, so the approximated GP posterior mean and variance are calculated in this paper. Similar to the work Dai et al., (2020), the posterior mean and variance are calculated according to formulae (6) and (7) in App .B of Dai et al., 2020. The corresponding computing formula for our ZGS-ED-DLCB in the DBO setting will be shown in our revised manuscript.
>
> (6). In response to your concerns, we have revisited our manuscript with a fresh perspective and revised it. If you are interested, the final version will deliever the revised manuscript.

---

### Meta-Review · Area_Chair_6ZoM · 2023-12-07

**Metareview:**

The authors consider the use of Bayesian optimization in a fully distributed multi-agent system. The authors propose an algorithm for this setting, analyze it theoretically, and evaluate it empirically.

The reviewers noted some interest in the problem setting of distributed Bayesian optimization, which perhaps has not been thoroughly explored by the existing literature in this area.

However, the reviewers noted some perceived weaknesses in the manuscript as submitted, some of which were judged to be severe:

- a lack of clarity in the presented material, with several reviewers expressing difficulty in understanding core developments
- as a result of the above, insufficient motivation for some decisions made by the authors in developing their methodology
- some concerns regarding the theoretical foundation and analysis

**Justification For Why Not Higher Score:**

Despite engagement with the authors during the author response period, 3 of 4 reviewers continued to express a perceived severe lack of clarity regarding the core ideas and developments in this paper. There was consensus that the manuscript would require significant revision before it would be acceptable for publication.

**Justification For Why Not Lower Score:**

N/A

---

### Decision · Program_Chairs · 2024-01-16

Reject